# Think Twice to See More: Iterative Visual Reasoning in Medical VLMs

## Abstract

Medical vision-language models (VLMs) excel at image-text understanding but typically rely on a single-pass reasoning that neglects localized visual cues. In clinical practice, however, human experts iteratively scan, focus, and refine the regions of interest before reaching a final diagnosis. To narrow this machine-human perception gap, we introduce ViTAR, a novel VLM framework that emulates the iterative reasoning process of human experts through a cognitive chain of "think-act-rethink-answer". ViTAR treats medical images as interactive cognitive objects, enabling models to engage multi-step visual reasoning. To support this approach, we curate a high-quality instruction dataset comprising 1K interactive examples that encode expert-like diagnostic behaviors. In addition, a 16K visual question answering training data has been curated towards fine-grained visual diagnosis. We introduce a two-stage training strategy that begins with supervised fine-tuning to guide cognitive trajectories, followed by the reinforcement learning to optimize decision-making. Extensive evaluations demonstrate that ViTAR outperforms strong state-of-the-art models. Visual attention analysis reveals that from the "think" to "rethink" rounds, ViTAR increasingly anchors visual grounding to clinically critical regions and maintains high attention allocation to visual tokens during reasoning, providing mechanistic insight into its improved performance. These findings demonstrate that embedding expert-style iterative thinking chains into VLMs enhances both performance and trustworthiness of medical AI systems. Codes can be available from the anonymous repository.

## 1 Introduction

Medical vision-language models (VLMs) have evolved from task-specific architectures to versatile frameworks, advancing large-scale medical image annotation (Xie et al., 2024), outcome prediction (Zhong et al., 2025), and clinical reasoning (Chen et al., 2024a). Powered by large language models (LLMs), systems such as LLaVA-Med (Li et al., 2023) and Lingshu (Xu et al., 2025) can engage human-like clinical dialogues and act as visual assistants. Nevertheless, current VLMs typically perform a single-pass inference strategy (Zhang et al., 2024), generating predictions from the entire images without explicitly identifying key visual cues that is vital for decision-making.

In the realm of medical imaging diagnosis, human experts follow an iterative cognitive process essentially comprising a multiscale observation (Aggarwal et al., 2021). Clinicians begin with a global image examination to locate suspicious regions of interest (ROIs). They always rely on the back-and-forth ROI-level reasoning to reach a final diagnostic conclusion (Shen et al., 2017). Therefore, equipping VLMs with analogous iterative reasoning is essential for cross-modality learning, yet current models remain largely confined to static image–question–answer training regimes (Zhang et al., 2023). While human-guided annotation can remarkably drive VLM's attention and enhance image interpretation (Xie et al., 2024; Shui et al., 2025), precise lesion-level annotations are costly to acquire at scale, especially in scenarios involving multi-round image-language dialogues. More critically, a fundamental challenge is to develop model frameworks to interact with medical images, refine their focus, and mimic human-like diagnostic reasoning. Without these abilities, VLMs are prone to overlook fine-grained abnormalities, break image-associated reasoning chains, and produce hallucinated interpretations.

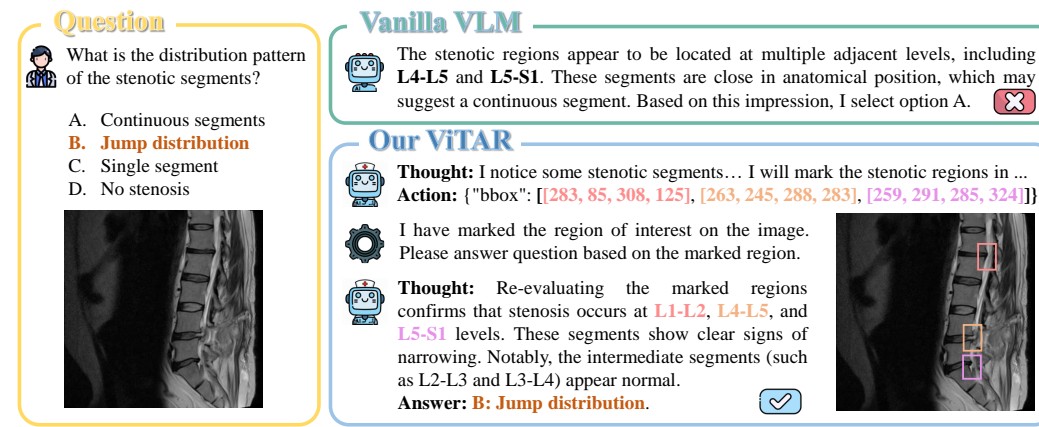

Figure 1: Comparison between the vanilla VLM and ViTAR. ViTAR specializes in explicit visual grounding with iterative reasoning. ViTAR first observes the image input, initiates an action to highlight key regions, and reasons over these regions to reach the final conclusion.

In this study, we propose **ViTAR** for **Vi**sual **T**hinking and **A**ction-centric **R**easoning. Our primary objective is to align the model's reasoning and action capability with clinician's iterative workflow. Unlike tool use (Li et al., 2024) and expert knowledge (Nath et al., 2025), we explore the iterative reasoning chains directly derived from the inherent learning ability of the model. In Figure 1, ViTAR incorporates structured reasoning with visual grounding in a "*think-act-rethink-answer*" cognitive process. Specifically, ViTAR first observes the image and engages in initial reasoning, then executes an action to mark the image based on the grounding parameters. Guided by the highlighted ROI regions, ViTAR refines its reasoning to generate the final prediction.

Conventional static datasets fail to capture the iterative thinking processes around the medical objects. To address this challenge, we curate a high-quality instruction dataset that encodes cognitive trajectories to mimic expert-like diagnostic behaviors for use in supervised fine-tuning (SFT). Simultaneously, we develop an automated pipeline to construct a large-scale visual question answering (VQA) corpus for reinforcement learning (RL). Through the paradigm of "*think-act-rethink-answer*", ViTAR delivers remarkable performance gains across multiple VQA benchmarks. Attention distribution analysis reveals a clear shift in attention patterns between the "think" and "rethink" rounds. We see that visual grounding attention becomes more tightly aligned with clinically critical regions. Furthermore, ViTAR allocates a high attention to visual tokens throughout the reasoning process, mitigating the "visual information diminishing" phenomenon often observed in conventional reasoning VLMs (Liu et al., 2025). Our main contributions are summarized as follows:

- We propose ViTAR, a novel VLM framework based on the *"think–act–rethink–answer"* paradigm, explicitly modeling multi-step visual reasoning strongly aligned with expert diagnostic workflows. ViTAR achieves substantial performance gains across multiple medical VQA benchmarks.
- We curate a 1K high-quality medical instruction dataset that reflects expert-level diagnostic trajectories, along with a 16K closed-ended VQA dataset for reinforcement learning, addressing the demand of data shortage in medical VLM studies.
- We provide mechanistic insights into improving multimodal reasoning ability. ViTAR's advance is supported by emerging capacities that enable dynamic adjustment of visual grounding attention and sustain high focus on visual tokens throughout the reasoning process.

## 2 RELATED WORK

### 2.1 MEDICAL VLMS

VLMs are strongly capable of handling multimodal healthcare data (Zhang et al., 2024). Representative models include Med-Flamingo (Moor et al., 2023), LLaVA-Med (Li et al., 2023), HuatuoGPT-Vision (Chen et al., 2024a), and Lingshu (Xu et al., 2025). These models leverage global image-text

pairs for cross-modal alignment and multimodal supervision. To identify fine-grained features, a structured region-of-interest (ROI) supervision (Liu et al., 2024a) is necessary. For example, R-LLaVA (Chen et al., 2024b) and MedTrinity (Xie et al., 2024) integrate ROI-based cues to guide visual encoding toward critical areas. fVLM (Shui et al., 2025) focuses on anatomical-level CT understanding to boost regional recognition. Collectively, these efforts drive a paradigm shift from global perception to region-focused modeling. However, relying on pre-defined annotations and single-step forward reasoning can not replicate the iterative processes in real-world settings. To address this hurdle, using external tools (Li et al., 2024) or domain-specific expert knowledge (Nath et al., 2025) could add fine-grained information. For instance, MMedAgent (Li et al., 2024) uses medical tools to handle various medical tasks and VILA-M3 (Nath et al., 2025) injects expert model knowledge in a static perception enhancement. Moving beyond external resource dependency, we explore the model's inherent ability to interact with the regions of interest, adjust its dynamics focus, and emulate expert-like iterative reasoning.

Recent advances in rule-based reinforcement learning (RL) can train medical VLMs to promote strong reasoning ability. Notable examples are Med-R1 (Lai et al., 2025) and MedVLM-R1 (Pan et al., 2025) that utilize structured reward signals to optimize the model's reasoning trajectory in medical VQA tasks. MedCCO (Rui et al., 2025) introduces the curriculum RL to process complex reasoning tasks. Chiron-o1 (Sun et al., 2025) proposes a mentor-intern collaborative search and adopts a progressive curriculum learning to advance VLM's reasoning capabilities. Nevertheless, these methods primarily target reasoning optimization without considering interactive visual exploration, which is a critical component of clinical decision-making. In addition, RL-based reasoning on the fine-grained perception remains an open challenge towards achieving expert-level cognitive capabilities in medical VLMs.

## 2.2 Thinking with Images

Despite the progress of VLMs and their variants, visual perception mechanism predominantly remains static (Chen et al., 2024a; Xu et al., 2025). For example, PeBR-R1 (Chen et al., 2025) produces single-round outputs and does not support visual interactive. To fully capture the chain of thinking in medical image diagnosis, models must enable interaction, action, and structured cognition towards fine-grained visual reasoning. Emerging "thinking with images" paradigm (Su et al., 2025c) transforms images from passive inputs into interactive cognitive workspaces (Wang et al., 2025a; Jiang et al., 2025). For instance, Pixel Reasoner (Su et al., 2025a) equips the model with zooming and region selection to enable a step-by-step visual thinking. OpenThinkIMG (Su et al., 2025b) integrates standardized visual interfaces with reinforcement learning in complex chart-based reasoning tasks. DeepEyes (Zheng et al., 2025) allows a mutual-information-driven tool selection to learn visual tool-based reasoning trajectories. ViGoRL (Sarch et al., 2025) achieves multi-step RL via complex multi-stage training, including Monte Carlo tree search, linearize, SFT and RL, to produce spatially grounded reasoning steps. SIMTOM (Wilf et al., 2024), although employing a two-stage prompting scheme, remains a prompt-engineered LLM without SFT or RL training, thus lacking the iterative perception–action–reflection loop needed for dynamic visual reasoning. VGR (Wang et al., 2025b), by contrast, relies on extensive chain-of-thought SFT with 158K region annotated samples in the general domain, whereas our method employs RL on a much smaller 16K medical dataset, enabling multi-round visual reasoning to emerge intrinsically. While showing the feasibility of interactive visual reasoning, these approaches often rely on predefined tool sets, domain-specific interfaces, or external expert knowledge. Their visual reasoning chains are largely mediated by external modules rather than emerging from the model's intrinsic learned capabilities. Moreover, the mechanisms underlying the iterative visual cue processing and reasoning remain insufficiently understood.

## 3 Method

### 3.1 ViTAR Framework

In Figure 2, ViTAR represents a VLM endowed with a "think-act-rethink-answer" cognitive capabilities to emulate the dynamic and interactive process in real-world clinical scenarios. We formalize the learning task as a multi-turn decision-making process. Given an initial medical image and a

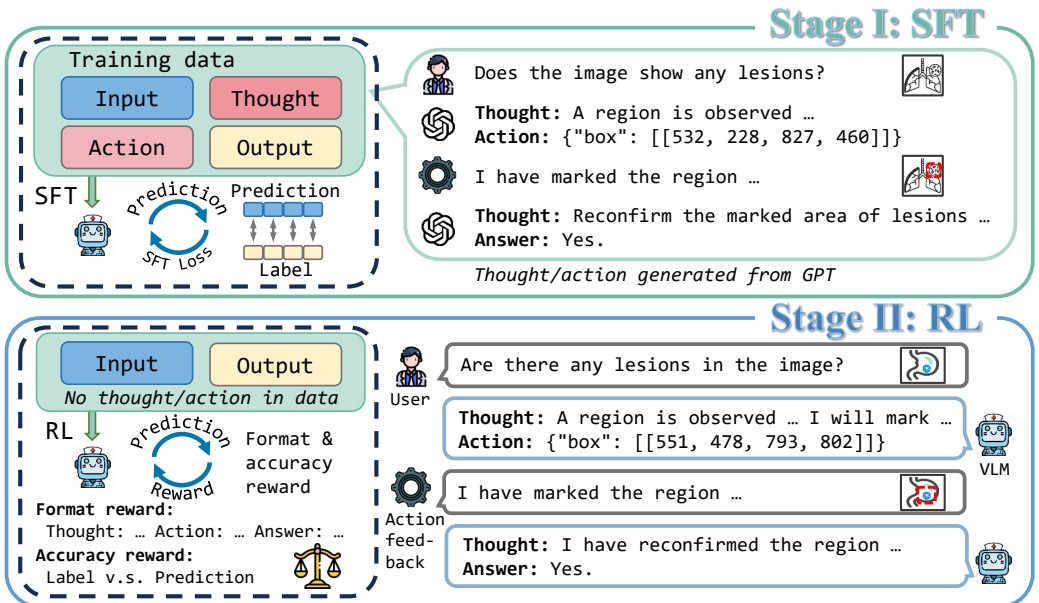

Figure 2: ViTAR's framework of visual thinking and action-centric reasoning. In supervised fine-tuning, ViTAR is trained with structured instructions to mimic expert-like reasoning patterns and region-marking behaviors. In Stage II, ViTAR is further optimized with rewards by reinforcement learning, shifting from imitation to autonomous decision refinement.

natural language query, the objective is to progressively think and act, receive feedback, and subsequently rethink, and ultimately generate an answer.

We adopt a two-stage, guide-and-optimize training strategy to enable multi-turn interactive capabilities in the VLM. In the first stage, supervised fine-tuning (SFT) is used to establish foundational "think–act–rethink–answer" patterns. The training objective of this stage is not to directly optimize the final answer but to guide the model to learn expert-style cognitive trajectories and decision-making policies, enabling structured perception and reasoning abilities for subsequent interactive stages. In the second stage, reinforcement learning (RL) is introduced to allow the model to optimize its reasoning ability based on reward signals, focusing on the answer accuracy and format precision.

## 3.2 TRAINING STRATEGY

### 3.2.1 STAGE I: SUPERVISED FINE-TUNING

In the first stage, we adopt a SFT strategy that familiarizes the model $\mathcal{M}$ to the step-by-step visual reasoning through a process of interleaved thought generation and action execution. The model is trained to maximize the likelihood of generating each target sequence in the correct order. The "think–act–rethink–answer" pattern consists of two rounds of dialogue. Specifically, in the first round of dialogue ($t = 0$), the model is taught to generate an initial thought $r_0$ and an associated action $a_0$ conditioned only on the input image $I$ and question $Q$. After the action is executed, corresponding feedback $f$ together with the updated image $I'$ are obtained. In the second round of dialogue ($t = 1$), the model subsequently re-evaluates its reasoning to produce a refined thought $r_1$ and the final answer $O$. Formally, the conditional probability of producing a target token sequence $y^{[t]}$ of length $L_t$ is decomposed auto-regressively as:

$$p_\theta\left(y^{[t]} \mid x^{(I)}, x^{(Q)}, \mathcal{C}_t\right) = \prod_{i=1}^{L_t} p_\theta\left(y_i^{[t]} \mid x^{(I)}, x^{(Q)}, \mathcal{C}_t, y_{<i}^{[t]}\right), \qquad (1)$$

where $\theta$ denotes trainable parameters in model $\mathcal{M}$, $\mathcal{C}_t$ represents all preceding context tokens, and $y^{[t]}$ is the generated token sequence in current step $t$. Here, $x^{(I)}$ and $x^{(Q)}$ denote the tokenized

representations of the image $I$ and question $Q$, respectively. In the first round of dialogue, the contextual tokens $\mathcal{C}_t$ are empty and $y^{[t]}$ denotes the generated tokens of $r_0$ and $a_0$, i.e., $y^{[t]} = (y^{(r_0)}, y^{(a_0)})$. In the second round, $\mathcal{C}_t$ comprises the token of the historical interaction information $(r_0, a_0, I', f)$, and $y^{[t]}$ denotes the generated tokens of $r_1$ and $O$, i.e., $y^{[t]} = (y^{(r_1)}, y^{(O)})$.

### 3.2.2 STAGE II: REINFORCEMENT LEARNING

After the Stage I training, the model learns to mimic the designed visual reasoning process, but can not make a reasoning with its own internal thinking. To facilitate the model's intrinsic reasoning, we introduce the stage of training with RL. We model the two-turn interactive medical reasoning task as a Markov Decision Process (MDP) (Puterman, 2014) with two steps. In each time step $t$, the decision consists of generating an intermediate reason followed by an action or the terminal answer. The state $S$ retains the complete history of reasoning outputs and actions to satisfy the Markov property.

For RL algorithm, we employ the Group Relative Policy Optimization (GRPO) (Shao et al., 2024) algorithm to enhance the model's autonomous decision-making capability within the dynamic cognitive loop. The VLM policy $\pi_\theta$ shares the same parameters $\theta$ of $\mathcal{M}$, optimizing the policy model directly results in optimization of the $\mathcal{M}$ as a whole. In the first round ($t = 0$), the initial state is $S_0 = (I, Q)$, where $I$ is the initial medical image and $Q$ is the query in natural language. The VLM policy $\pi_\theta$ produces an intermediate thought $r_0$ and an action $a_0$ given $S_0$, i.e., $(r_0, a_0) \sim \pi_\theta(\cdot \mid S_0)$. The action $a_0$ triggers region annotation as visual interactions. The environment $\mathcal{E}$ applies the action to the image and returns an updated image $I'$ along with feedback $f$, i.e., $(I', f) = \mathcal{E}(I, a_0)$.

In the second round ($t = 1$), the state is $S_1 = (I, Q, r_0, a_0, I', f)$, which encodes the initial inputs and the complete result of the first step. Based on $S_1$, the VLM policy generates a second thought $r_1$ and the final output $O$, i.e., $(r_1, O) \sim \pi_\theta(\cdot \mid S_1)$.

For the design of RL rewards, we incorporate a format reward and an accuracy reward. Specifically, a format reward $R_{\text{format}}$ is introduced to encourage the generation of standardized outputs: the model receives 0.2 points if the "thought" and "action" fields can be successfully parsed according to the prescribed schema (e.g., `{"thought":"the reasoning process", "actions":[{"name":"Mark","arguments":[[89,85,146,145]]}]}`), and an additional 0.2 points if the final answer adheres to the expected format (e.g., `{"actions":[{"name":"Terminate","arguments":{"answer":"A"}}]}`). To further encourage the model to produce correct answers, we employ an accuracy reward $R_{\text{acc}}$, whereby the model receives a score of 1.0 if the final answer exactly matches the ground-truth label, and 0.0 otherwise. The overall reward is thus defined as: $R = R_{\text{format}} + R_{\text{acc}}$.

## 4 DATASET CURATION

Training data quality and scale remain a bottleneck for developing reliable medical VLMs (Xu et al., 2025). A wide range of medical VQA datasets (Hu et al., 2024; Chen et al., 2024a) adopt a static image-question-answer triplet structure. These static datasets are unable to provide fine-grained lesion annotations with dynamic cognitive trajectories in clinical reasoning. To overcome this hurdle, we propose a systematic data construction method that leverages the medical object detection datasets to generate high-quality VQA training samples. In particular, these data samples support fine-grained VQA over local regions and enable dynamic interactive reasoning. This curation pipeline has two core components: (1) an automatic question-answer generation pipeline based on the detection data, and (2) an instruction-based data augmentation mechanism designed to produce interactive cognitive chains.

### 4.1 VQA CONSTRUCTION FROM DETECTION DATA

**VQA Generation** We begin VQA generation by selecting well-structured medical object detection datasets from Roboflow (Robicheaux et al., 2025). We extract detection category labels and corresponding bounding box coordinates as the fundamental input for the VQA generation. For each detected category, we design diverse question templates. The designed question templates cover presence or absence detection of specific findings, spatial location determination, lesion recognition

at specified coordinates, and other related clinically relevant query types. We then use an LLM combined with the detection information and question templates to generate semantically meaningful and contextually relevant questions and answers. This enables an effective conversion of object detection data into VQA data. A representative prompt including diverse templates is shown in Appendix Figure 17. The specific templates in prompt are adapted according to the characteristics and domain requirements of each detection dataset.

**VQA Validation**   To ensure the utility of the generated VQA pairs for model training, we design an LLM-based validation pipeline. In this process, the LLM receives the input including image resolution, category label, and bounding box coordinates, from which it generates a prediction. If the prediction is inconsistent with the provided answer, the sample is flagged as semantically ambiguous and discarded. This mechanism can mitigate errors in the generated QA pairs caused by template deficiencies or incomplete target detection data, thereby remarkably enhancing the overall quality and reliability of the dataset. A prompt for verifying the VQA is provided in Appendix Figure 18.

We utilize Qwen2.5-72B-Instruct (Yang et al., 2024) to generate and verify VQA samples in our constructed training dataset. Based on 19 curated object detection datasets and 69 task-specific templates, we construct this high-quality VQA corpus comprising 16,601 samples, which involves tasks with diverse imaging modalities, lesion types, and cognitive complexities. See the Appendix Section A.4 for more information of constructed VQA. Examples of our constructed VQA data are presented in Appendix Figures 12 and 13.

### 4.2   INTERACTION INSTRUCTION GENERATION

Conventional VQA corpora are static triplets that can not reflect the iterative observations and operations. In our study, we seek to perform iterative reasoning and actions around VLMs. For instance, we ask VLMs to accomplish actions such as draw bounding boxes around lesions. To this end, we propose an instruction construction scheme that simulates the cognitive processes in clinical image settings. Based on object detection annotations, the model is guided to construct dynamically structured QA chains, reflecting the expert decision-making loop. The scheme involves the two stages: **(1) Initial thinking and action**. We employ the LLM that takes as input the bounding box, category information and QA. Based on this input, the LLM generates an initial thought and an intended action (e.g., "mark the region with a bounding box"). **(2) Re-thinking after action.** We use the LLM to provide a detailed diagnostic analysis and produce a final decision by assuming the proposed action has been carried out.

We leverage GPT-4o (Hurst et al., 2024) to produce thoughts (detail prompt is provided in Appendix Figure 16). By systematically combining generated initial thinking, actions, and re-thinking into VQA data, we ultimately obtain 1,000 detailed interactive instructions. We observe that these instructions evidently enhance the interpretability of the VQA corpus, enabling the model to learn a multi-turn reasoning process aligned with human-expert's cognition. Examples of our constructed interaction instructions are presented in Appendix Figures 10 and 11.

## 5   EXPERIMENTS

### 5.1   SETTING

**Benchmarks**   To comprehensively evaluate the action-centric reasoning performance, we conduct experiments on seven representative datasets. PathVQA (He et al., 2020), SLAKE (Liu et al., 2021), and VQA-RAD (Lau et al., 2018) are widely used benchmarks in medical VQA research. Omni-iMedVQA (Hu et al., 2024) focuses on medical image perception tasks by constructing multiple classification datasets into QA form. PMC-VQA (Zhang et al., 2023) is derived from PubMed Central that contains 2,000 medical QA pairs annotated by human experts. Towards a higher-level reasoning, MMMU-Med (Yue et al., 2024) and MedXpertQA (Zuo et al., 2025) offer challenging QA scenarios. MMMU-Med is a medical subset of health and medicine extracted from the popular multimodal reasoning benchmark MMMU (Yue et al., 2024). MedXpertQA presents a more difficult setting, simulating real clinical licensing exams to assess whether models can perform medical reasoning and decision-making at a near-expert level.

| Method | PMC. | Path. | SLAKE | RAD. | Omni. | MMMU | MedX. | Ave. |
|--------|------|-------|-------|------|-------|------|-------|------|
| *Proprietary models* | | | | | | | | |
| GPT-4.1 | 55.2 | 55.5 | 72.2 | 65.0 | 75.5 | 75.2 | 45.2 | 63.4 |
| Claude-Sonnet-4 | 54.4 | 54.2 | 70.6 | 67.6 | 65.5 | 74.6 | 43.3 | 61.5 |
| Gemini-2.5-Flash | 55.5 | 55.4 | 75.8 | 68.5 | 71.0 | 76.9 | 52.8 | 65.1 |
| *General-purpose VLMs* | | | | | | | | |
| LLaVA-v1.5-8B | 36.4 | 54.1 | 59.4 | 54.2 | 48.8 | 38.2 | - | 48.5 |
| LLaVA-Next-7B | 35.5 | 47.9 | 57.9 | 52.6 | 49.6 | 33.1 | 20.7 | 42.5 |
| LLaVA-Next-13B | 36.6 | 51.9 | 58.9 | 55.8 | 50.6 | 39.3 | 19.1 | 44.6 |
| Qwen2.5-VL-7B | 50.4 | 63.9 | 69.5 | 64.9 | 56.5 | 56.7 | 21.7 | 54.8 |
| InternVL3-8B | 52.7 | 65.3 | 76.9 | 67.3 | 72.3 | 64.7 | 22.3 | 60.2 |
| *Medical-specific non-reasoning VLMs* | | | | | | | | |
| Med-Flamingo | 23.3 | 54.7 | 43.5 | 45.4 | 30.2 | 28.3 | 19.3 | 35.0 |
| RadFM | 25.9 | 38.7 | 34.6 | 50.6 | 30.5 | 27.0 | 19.8 | 32.4 |
| LLaVA-Med-7B | 24.7 | 56.8 | 48.6 | 51.4 | 44.5 | 36.9 | 20.3 | 40.5 |
| HuatuoGPT-V-8B | 51.4 | 59.8 | 66.8 | 59.4 | 65.4 | 56.7 | 21.6 | 54.4 |
| Lingshu-7B | 55.6 | **71.7** | 78.9 | 62.2 | 71.7 | 70.0 | 24.8 | 62.1 |
| *Medical-specific reasoning VLMs* | | | | | | | | |
| Med-R1 | 45.8 | 53.3 | 55.1 | 55.9 | - | 32.7 | 20.3 | 43.9 |
| MedVLM-R1 | 44.8 | 55.2 | 65.9 | 61.4 | - | 35.5 | 21.2 | 47.3 |
| Chiron-o1-8B | 57.5 | 74.0 | 83.2 | 76.8 | - | 54.6 | 23.8 | 61.7 |
| MedCCO-7B | 53.2 | 82.8 | 79.4 | 76.3 | 65.8 | 59.3 | 23.2 | 62.9 |
| **ViTAR** | **57.2** | 67.0 | **80.8** | 70.1 | 74.2 | **72.0** | **26.9** | 64.0 |
| **ViTAR (w/ IoU)** | 56.2 | 69.6 | **80.8** | 71.7 | **74.7** | 72.0 | 25.7 | **64.4** |

Table 1: Performance comparison of different categories VLMs on seven medical VQA benchmarks. Overall, ViTAR achieves the leading performance on most benchmarks, highlighting the effectiveness of interactive reasoning. Bold numbers indicate the best result in open-source VLMs and gray numbers indicate that the model has been trained on the corresponding dataset.

**Comparison and Implementation**  We compare ViTAR with representative models. Proprietary models include GPT-4.1 (OpenAI, 2025), Claude-Sonnet-4 (Anthropic, 2025) and Gemini-2.5-Flash (Comanici et al., 2025). General-purpose models include LLaVA-v1.5-8B (Liu et al., 2023), LLaVA-Next-7B (Liu et al., 2024b), LLaVA-Next-13B, Qwen2.5-VL-7B (Bai et al., 2025) and InternVL3-8B (Zhu et al., 2025), representing the mainstream architectures for open-domain multimodal understanding. Medical-specific non-reasoning baselines include Med-Flamingo (Moor et al., 2023), RadFM (Wu et al., 2023), LLaVA-Med-7B (Li et al., 2023), HuatuoGPT-Vision-8B (Chen et al., 2024a), and Lingshu-7B (Xu et al., 2025), all of which have shown strong performance in medical image understanding. Medical-specific reasoning VLM include Med-R1 (Lai et al., 2025), MedVLM-R1 (Pan et al., 2025), Chiron-o1-8B (Sun et al., 2025) and MedCCO-7B (Rui et al., 2025). Lingshu-7B serves as both the foundation model for our work and the main baseline in our experiments. Our model is trained on a single NVIDIA A800 node (8 × 80G GPUs) and the training is completed within 23 hours. Detailed setups for each sub-experiment are provided in Appendix Section A.2. Prompts used for testing are presented in the Appendix Figure 20.

## 5.2 EXPERIMENTAL RESULTS

**Main Results**  In Table 1, ViTAR demonstrates its robust generalization in basic visual perception and vision-language reasoning on seven medical benchmarks. Against non-reasoning medical VLMs, ViTAR leads in performance on most benchmarks. On VQA-RAD, ViTAR surpasses all of these models and surpasses the baseline model of Lingshu (78.9%) by nearly 8 points. On PathVQA, we note a performance degradation relative to Lingshu. This is likely due to two factors: (i) Lingshu is trained on the PathVQA dataset (Xu et al., 2025) whereas ViTAR is trained on data that exclude PathVQA; (ii) Our dataset has limited pathology-related samples as seen in Appendix Figure 9. Against reasoning-enabled VLMs, ViTAR substantially outperforms Med-R1 and MedVLM-R1.

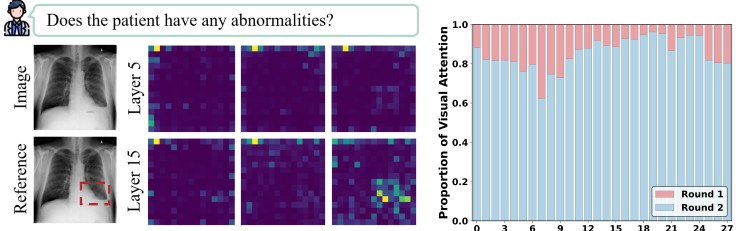

(a) Visual grounding attention comparison between Lingshu and our ViTAR in round 1 and round 2.

(b) Visual attention allocation comparison in round 1 and round 2 of our ViTAR.

(c) Visual attention allocation between Lingshu and ViTAR.

Figure 3: Compared to the first "think" statue (round 1), the second "rethink" statue (round 2) achieves more precise alignment with annotated lesion regions and allocates a higher proportion of attention to visual tokens. And ViTAR outperforms Lingshu by sustaining focused attention on verifiable regions and allocating more visual attention. See Appendix Figure 8 for more details.

Since Chiron-o1 and MedCCO are already trained on portions of these benchmarks, we exclude them from direct performance comparisons and highlight them in gray.

On two reasoning-intensive benchmarks, MMMU-Med and MedXpertQA, ViTAR achieves 72.0% on MMMU-Med, surpassing both medical-specific non-reasoning and reasoning VLM, and closely matching certain proprietary models, reflecting its enhanced ability in multi-turn vision-language reasoning. On the most challenging MedXpertQA, ViTAR achieves a score of 26.9%, consistently outperforming all other open-source models with comparable parameter scale. While proprietary models maintain a lead, ViTAR's performance is achieved using only a 7B parameter scale, which is significantly smaller than proprietary counterparts, indicating considerable potential for parameter-efficient medical reasoning.

**ViTAR's Intrinsic Reasoning Ability**   To investigate whether ViTAR's intrinsic reasoning capability can surpass the performance limits of external tool-assisted approaches, we conduct experiments on the SLAKE dataset, which provides the required organ or lesion annotations. In Figure 4, ViTAR reaches 80.77% without relying on any external tool, even surpassing Lingshu's performance with human annotations as the upper bound of external detection tools. When further combined with human annotations, ViTAR's performance rises to 81.97%, revealing that its inherent capability is orthogonal to the gains from the external assistance, enabling a complementary performance gain. We additionally conduct the experiment with using segmentation masks instead of bounding boxes during inference. We observe a notable performance gain that ViTAR achieved 83.41 with segmentation masks, compared to 81.97 using bounding boxes. Overall, these results highlight ViTAR's advances in its strong intrinsic reasoning capacity and integration of external signals. See Appendix Section A.2 for more experimental details.

**Why *"think–act–rethink–answer"* cognitive processes enhance reasoning?**   To gain insights into the multi-round visual cognitive process, we analyze the differences between the "think" (round 1) and "rethink" (round 2) in ViTAR from perspectives of the visual grounding and visual attention allocation. Our results show that the second round achieves substantially better visual grounding with attention more sharply focused on crucial regions (Figure 3a). We extend the comparison to the baseline Lingshu (Xu et al., 2025). Lingshu fails to consistently align with annotated lesion regions. ViTAR, by contrast, highlights key regions closely matching the reference annotations.

Visual attention allocation analysis shows that ViTAR allocates a higher proportion of attention to visual tokens during the second round compared to the first round (Figure 3b). Its overall visual attention allocation also exceeds that of Lingshu (Figure 3c), mitigating a "visual information diminishing" and its consequential hallucination  (Liu et al., 2025) often seen in conventional reasoning VLM. Taken together, these results reveal that multi-round thinking represents a genuine refinement mechanism rather than mere repetition. Second-round reasoning sharpens visual grounding and attention, redirecting the focus from language-driven associations towards verifiable visual evidence, thereby driving consistent performance improvements. See Appendix Figure 8 for more details of visual attention allocation.

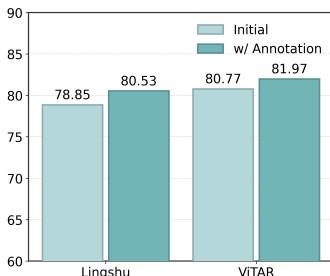

Figure 4: Performance comparison with human annotation.

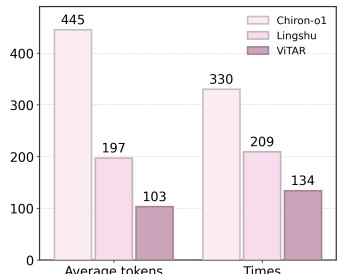

Figure 5: Comparison with reasoning efficiency (Times: S).

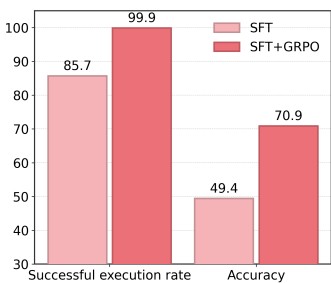

Figure 6: Comparison with SFT and GRPO.

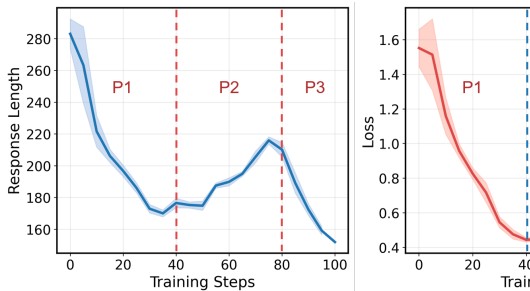

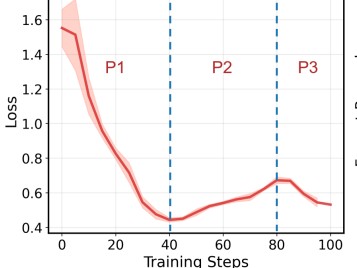

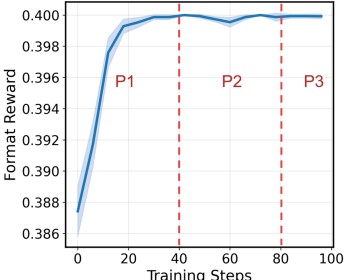

Figure 7: Evolution of RL training over three phases. Shading indicates confidence interval.

**Step-wise Reasoning Consistency and Attention–ROI Overlap**   We use GPT-5 to evaluate the semantic consistency between the think and rethink stages. ViTAR achieves 98.6% step-wise consistency, indicating that intermediate reasoning steps are stable. To assess whether the model grounds its reasoning in meaningful areas, we compute visual attention allocations inside vs. outside annotated ROIs in Table 2. For ViTAR, the average attention score inside ROIs is 0.2576. 68.9% of the samples exhibit higher average attention inside the ROI than outside, showing that the model's spatial focus aligns well with medically relevant areas. In contrast, for the baseline model, the average attention score inside ROIs is only 0.1033, and the proportion of samples with higher in-ROI attention is just 33.6%. These results collectively provide strong evidence that ViTAR's multi-step reasoning is faithful, and that the model's attention is meaningfully guided by relevant regions.

**Human Evaluation of Iterative Reasoning**   We conduct a human evaluation on a random subset of 100 samples, assessing each along three dimensions: reasoning clarity and logical coherence, observation completeness and alignment, and answer correctness. Human evaluators are allowed to consult additional resources, including LLM and web searches, to verify response accuracy. Results show that ViTAR outperforms a strong single-pass baseline in 49 cases, while the baseline outperforms ViTAR in 34 cases, with the remaining 17 cases resulting in ties where both models produce incorrect answers. These findings suggest that ViTAR's iterative reasoning process enhances the completeness of intermediate reasoning, improves human trust and agreement, and supports our claim that ViTAR narrows the machine-human perception gap compared to single-pass VLMs.

**Impact of the Weights on Format Rewards**   We conduct experiments with different format rewards to evaluate their influence on medical VQA performance. The results show that overall performance remains highly stable across reward values of 0.1, 0.2, and 0.4, with average scores consistently around 64 in Table 3. We further explore the effect of adding an IoU-based reward to promote spatial alignment during training. Relative to the w/o IoU setting, incorporating the IoU reward yields slight yet consistent improvements across multiple datasets, increasing the overall average from 64.0 to 64.4 in Table 1. This suggests that the IoU reward provides a modest but beneficial training signal that strengthens spatial grounding while preserving overall stability.

| Model | Avg. In-ROI | Avg. Out-ROI | % Samples w/ Higher In-ROI |
|---|---|---|---|
| Baseline | 0.1033 | 0.0863 | 33.6% |
| ViTAR | 0.2576 | 0.1237 | 68.9% |

Table 2: Comparison of visual attention allocation inside vs. outside annotated ROIs.

| Reward | PMC-VQA | Path. | SLAKE | VQA. | Omni. | MMMU | MedX. | Ave. |
|---|---|---|---|---|---|---|---|---|
| 0.1 | 55.9 | 71.5 | 79.8 | 70.5 | 72.7 | 72.7 | 24.8 | 64.0 |
| 0.2 | 58.7 | 69.9 | 81.7 | 71.3 | 71.4 | 72.7 | 23.7 | 64.2 |
| 0.4 | 57.2 | 67.0 | 80.8 | 70.1 | 74.2 | 72.0 | 26.9 | 64.0 |

Table 3: Performance comparison under different format reward.

**RL from Scratch vs. SFT+RL** We conduct three independent RL from scratch trials, all of which failed within the first 10 RL steps. Specifically, the reward quickly collapsed to abnormal values. And model outputs degenerated into nonsensical or corrupted strings, causing the program to halt. This failure is expected given our task setting because our multi-turn framework requires strictly formatted outputs at each step. Both the first and second model responses must adhere to a predefined schema that allows the system to parse and execute actions correctly.

**Reasoning and Execution Efficiency** We conduct a systematic evaluation of reasoning efficiency on two medical reasoning tasks, MMMU-Med and MedXpertQA, measuring the average inference time and the mean number of generated tokens for three models: the reasoning model Chiron-o1, the non-reasoning model Lingshu, and our proposed model ViTAR. In Figure 5, ViTAR substantially reduces reasoning length and reasoning time while achieving the highest performance. This short reasoning length could underpin the high visual attention allocation. Despite adopting a two-round reasoning paradigm, ViTAR reaches a speedup of 2.46× over Chiron-o1 and 1.56× over Lingshu.

We evaluate the impact on action execution reliability and question-answering accuracy with or without RL training (GRPO). The action execution success rate measures whether the model correctly generates and parses actions. Results in Figure 6 show that introducing RL significantly improved the success rate from 85.7% to 99.9%, indicating substantial gains in execution stability and accuracy after RL.

We observe the evolving ability improvements during the training dynamics of RL, as seen in Figure 7. Three phases can be identified in the changing dynamics of response length, training loss, and format rewards. During P1, the model initially demonstrates a certain degree of action invocation capability, which significantly accelerated its adaptation to the action execution mechanism. During this phase, we observe a gradual decrease in response length and a steady increase in format reward, indicating that the model is beginning to learn how to generate structured and executable action calls by RL, which ensures the action execution reliability. In P2, the model enters a deeper phase of policy enhancement. It begins to generate longer reasoning paths and explore more complex cognitive processes. And the format reward approaches its saturation value. For P3, the model starts to compress its reasoning paths again, shifting towards more efficient task completion. Although the response length decreases further, the execution success rate remains stable, indicating that the model has learned to execute actions efficiently without redundant reasoning.

## 6 CONCLUSION

We present ViTAR as a vision-language model framework for enhancing medical visual reasoning. ViTAR treats medical images as interactive cognitive objects and emulates expert-like diagnostic behavior through a structured "think-act-rethink-answer" cognitive chain. We curate a high-quality instruction dataset capturing expert reasoning trajectories and a large-scale fine-grained VQA dataset for RL training. Extensive experiments demonstrate the superiority of ViTAR over competing methods across seven medical VQA benchmarks, supporting ViTAR's intrinsic and internalized visual thinking ability as well. In particular, we observe that during the "rethink" phase, ViTAR increasingly concentrates its visual grounding attention on critical regions while maintaining strong attention to visual tokens among numerous text tokens. These findings underscore the potential of embedding expert-style cognitive processes into VLMs, offering an opportunity towards expert-level and trustworthy AI systems in healthcare.

ETHICS STATEMENT

This work does not involve experiments with human subjects, animal testing, or personally identifiable information. All datasets used in this study are derived from publicly available sources. The datasets, methodologies, and results presented here do not pose foreseeable risks of harm. All contributions have been conducted in accordance with the ICLR Code of Ethics, with full compliance to guidelines.

REPRODUCIBILITY STATEMENT

We have made efforts to ensure that the results presented in this paper are fully reproducible. A complete description of our experimental setup, including hyperparameter configurations, model architectures, and training procedures, is provided in Section 5.1 and Appendix Sections A.2 and A.3. To facilitate reproducibility, we provide an anonymous repository to our source codes, together with instructions for running experiments.

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

# A APPENDIX

## A.1 ADDITIONAL EXPERIMENTS

**Experiments on OmniMedVQA**  Table 4 presents results on the OmniMedVQA benchmark, encompassing eight imaging modalities: computed tomography (CT), dermatology (Der), fundus photography (FP), magnetic resonance imaging (MRI), microscopy (Mic), optical coherence tomography (OCT), X-ray, and ultrasound (US). ViTAR attains the highest overall accuracy of 74.2%, outperforming all baselines, and delivers competitive scores in most modalities. The gains are particularly pronounced in Der (82.3%), FP (83.4%), OCT (84.2%), and US (84.9%). A modest degradation is observed in CT performance, which we attribute to a domain shift, because our CT training subset predominantly contains pulmonary nodule cases (Figure 9). Compared with the strongest general-purpose VLM, InternVL3-8B, ViTAR yields an absolute improvement of +1.9 percentage points in average accuracy and achieves a more balanced distribution of results across modalities. These findings demonstrate that the ViTAR architecture generalizes effectively to a wide spectrum of medical imaging types, sustaining robustness even in modality-diverse diagnostic environments.

**Experiments on MMMU-Med**  Table 5 reports the performance on the MMMU-Med benchmark, which covers five subcategories, including basic medical science (BMS), clinical medicine (CM), diagnostics and laboratory medicine (DLM), pharmacy (P), and public health (PH). ViTAR achieves the best overall performance with an average accuracy of 72.0%, surpassing Lingshu-7B by 2.0 percentage points. Within the medical-specific VLM group, Lingshu-7B ranks second at 70.0% average accuracy, while among general-purpose VLMs, InternVL3-8B leads with 64.7%. Notably, ViTAR surpasses InternVL3-8B by a substantial margin of +7.3 percentage points, indicating not only superior absolute accuracy but also more balanced performance across domains. These results demonstrate that ViTAR maintains strong generalization across heterogeneous medical disciplines and exhibits robust competence in both basic science questions and complex clinical scenarios.

**Sensitivity to Action Output Formats**  Since the format of action outputs can impact both execution accuracy and compatibility with downstream processing systems, we evaluate two alternative representations: an explicit structured format and a concise implicit format. An explicit command format clearly specifies both the action name and its arguments, following standard JSON syntax (e.g., `{"actions":[{"name": "Mark", "arguments":{"box":[[64,15,124,65]]}}]}`). In contrast, the implicit format provides a more compact representation (e.g., `<bbox>[[64, 15, 124, 65]]</bbox>`). In Table 6, experimental results indicate that the explicit format generally outperforms the implicit one, with improvements ranging from 0.5% to 3.7%. These findings highlight that while both action formats are compatible with the training paradigm, using an explicit structured action format adhering to the

| Method | CT | Der | FP | MRI | Mic | OCT | X-Ray | US | Average |
|---|---|---|---|---|---|---|---|---|---|
| *General-purpose VLM* | | | | | | | | | |
| LLaVA-v1.5-7B | 33.0 | 63.1 | 49.7 | 53.8 | 48.4 | 76.0 | 56.6 | 31.2 | 48.8 |
| LLaVA-NEXT-7B | 40.1 | 54.0 | 39.5 | 54.8 | 48.8 | 58.4 | 53.3 | 47.9 | 49.6 |
| LLaVA-NEXT-13B | 40.0 | 58.0 | 43.6 | 47.4 | 50.5 | 63.2 | 59.6 | 42.6 | 50.6 |
| Qwen2.5-VL-7B | 65.0 | 62.3 | 70.8 | 52.4 | 64.6 | 58.4 | 74.8 | 27.6 | 56.5 |
| InternVL3-8B | 71.5 | 65.5 | 76.4 | 69.3 | 80.4 | 77.0 | 83.9 | 69.7 | 72.3 |
| *Medical-specific VLM* | | | | | | | | | |
| Med-Flamingo | 34.6 | 28.3 | 33.3 | 27.5 | 28.1 | 26.0 | 30.1 | 33.2 | 30.2 |
| RadFM | 33.3 | 36.3 | 35.0 | 22.0 | 28.0 | 31.3 | 31.5 | 26.1 | 30.5 |
| LLaVA-Med-7B | 25.3 | 45.2 | 48.4 | 35.9 | 44.0 | 42.1 | 31.7 | 83.7 | 44.5 |
| HuatuoGPT-V-8B | 73.3 | 61.2 | 76.5 | 62.2 | 71.0 | 70.4 | 79.7 | 44.5 | 61.0 |
| Lingshu-7B | 74.2 | 78.0 | 78.0 | 64.5 | 74.9 | 81.8 | 77.6 | 72.6 | 71.7 |
| ViTAR (ours) | 64.0 | 82.3 | 83.4 | 68.6 | 78.4 | 84.2 | 81.0 | 84.9 | 74.2 |

Table 4: Evaluation results on eight modalities from the OmniMedVQA.

| Method | BMS | CM | DLM | P | PH | Average |
|---|---|---|---|---|---|---|
| *General-purpose VLM* | | | | | | |
| LLaVA-1.5-7B | 42.3 | 44.0 | 37.0 | 34.7 | 35.2 | 38.2 |
| LLaVA-Next-7B | 40.5 | 36.9 | 32.1 | 32.3 | 26.9 | 33.1 |
| LLaVA-Next-13B | 53.6 | 46.7 | 33.3 | 22.2 | 40.0 | 39.3 |
| Qwen2.5-VL-7B | 53.3 | 66.7 | 30.0 | 60.0 | 73.3 | 56.7 |
| InternVL3-8B | 63.3 | 70.0 | 40.0 | 63.3 | 86.7 | 64.7 |
| *Medical-specific VLM* | | | | | | |
| Med-Flamingo | 29.6 | 28.1 | 24.8 | 25.3 | 31.2 | 28.3 |
| RadFM | 27.5 | 26.8 | 25.8 | 24.7 | 29.1 | 27.0 |
| LLaVA-Med-7B | 39.9 | 39.1 | 34.6 | 37.4 | 34.0 | 36.9 |
| HuatuoGPT-V-8B | 61.0 | 58.8 | 50.0 | 44.7 | 38.7 | 49.1 |
| Lingshu-7B | 70.0 | 73.3 | 63.3 | 83.3 | 73.3 | 70.0 |
| ViTAR (ours) | 73.3 | 76.7 | 63.3 | 73.3 | 73.3 | 72.0 |

Table 5: Performance on five subcategories from the MMMU-Med.

| Format | PMC-VQA | PathVQA | SLAKE | VQA-RAD | OmniMed. | MMMU-Med |
|---|---|---|---|---|---|---|
| Implicit | 55.7 | 63.7 | 80.3 | 68.5 | 70.4 | 72.0 |
| Explicit | 57.2 | 67.0 | 80.8 | 70.1 | 74.2 | 72.0 |
| Δ | 1.5 | 3.3 | 0.5 | 1.6 | 3.7 | 0.0 |

Table 6: Comparison with structured explicit format and concise implicit format.



(a) Attention allocation of Lingshu.  (b) Attention allocation during the first "think" round of ViTAR.  (c) Attention allocation during the second "rethink" round of ViTAR.

Figure 8: Attention allocation in VLMs. ViTAR substantially increases visual attention from the first to the second round, far surpassing the baseline Lingshu and alleviating visual attention sparsity.

standard JSON syntax can enhance accuracy. We argue that the explicit structured action command format is more aligned with function-calling paradigms in LLM (He et al., 2024).

**Attention Allocation** We quantitatively evaluate the visual attention allocation in "think-act-rethink-answer" cognitive process, defined as the proportion of attention weights allocated to visual tokens. As shown in Figures 8b and 8c, the average ratio (over 0-15 layers) increases from 5.84% in the first "think" round to 26.12% in the second "rethink" round. ViTAR allocates 2.82× more attention to visual tokens than Lingshu (9.26%) seen in Figure 8a), reflecting enhanced visual attentions. The second round conditions on the initial answer. This step tends to retrieve and integrate additional visual information, which increases attention allocation to visual tokens. Such behavior aligns with the model's ability to verify and refine earlier reasoning. These results suggest that the combination of architectural and training optimizations with the multi-round reasoning paradigm mitigates the "visual attention diminishing" issue commonly observed in the long reasoning process of RL models. Consequently, ViTAR is able to leverage visual information more effectively, which is likely to improve the accuracy and robustness of its multimodal reasoning.

## A.2 SUPPLEMENTARY EXPERIMENT DETAILS

**Details of "Main Results" in Section 5.2**   We divide the general-purpose and medical-specific non-reasoning models into two groups. The first group contains post–January 2025 releases, including Qwen2.5-VL-7B (Bai et al., 2025), InternVL3-8B (Zhu et al., 2025), HuatuoGPT-Vision-8B-Qwen2.5 (HuatuoGPT-V-8B) (Chen et al., 2024a), and Lingshu-7B (Xu et al., 2025). These models are deployed with vLLM version 0.9.2 (Kwon et al., 2023) and evaluated under identical prompt (Figure 20) with a temperature of 0.1. The second group consists of models released before January 2025. Since these models exhibit lower performance than ViTAR and some do not support online deployment with vLLM, we adopt the official inference results reported by HuatuoGPT-Vision. Results for proprietary models are taken from the Lingshu report. As for the reasoning models, the results of Med-R1 (Lai et al., 2025), MedVLM-R1 (Pan et al., 2025), and Chiron-o1 (Sun et al., 2025) are reported as in the Chiron-o1 paper. And the results for MedCCO (Rui et al., 2025) are taken from the MedCCO publication. ViTAR performs inference through a multi-round dialogue. In the first round, the model generates a reasoning trace (thought) and an action based on the input image and query. The system annotates the image according to the parameters in the action plan. In the second round, the annotated image and the dialogue history from the initial round are provided to the model for further reasoning with a focus on target regions. The model produces a precise and interpretable answer.

**Details of "Intrinsic Reasoning in ViTAR" in Section 5.2**   The SLAKE dataset contains image–question pairs for VQA and provides organ or lesion annotations. In the **Lingshu w/ annotation** experiment, the original images are replaced with their bounding-box-annotated versions of SLAKE. This setting simulates the use of tool-assisted perception, where the Lingshu model is evaluated with external visual cues. In the **ViTAR w/ annotation** experiment, the replacement occurs during the second dialogue round. The annotated images generated by action are replaced with the bounding-box–annotated images from SLAKE. Apart from this modification, all steps follow the standard ViTAR inference pipeline.

**Details of "Why *'think–act–rethink–answer'* cognitive processes enhances reasoning" in Section 5.2**   We use the visualization implementation from paper (Liu et al., 2025) as the basis for our visual analysis. For the ViTAR model, we use specific system instructions that guide the execution of the "think–act-rethink-answer" process. All other models operate with their default system prompts.

**Details of "Sensitivity to Action Output Formats" in Section A.1**   We evaluate two distinct representations of action models in our experiments. The first is an explicit structured format, where actions are expressed in JSON with explicitly specified parameters. For examples, `{"actions":[{"name":"Mark","arguments":{"box":[[64,15,124,65]]}}]}`. The corresponding final answer is represented as: `{"actions":[{"name":"Terminate", "arguments":{"answer":"A"}}]}`. The second is a concise implicit format, where action parameters are provided using tag-based notation. For example, `<bbox>[[64, 15, 124, 65]]</bbox>`. The corresponding final answer is `<answer>A</answer>`.

## A.3 TRAINING DETAILS

**Stage I: Supervised Fine-tuning**   We perform SFT of the Lingshu-7B (Xu et al., 2025). Training uses the `Transformer Reinforcement Learning (TRL)` framework (von Werra et al., 2020) together with `DeepSpeed` (Rasley et al., 2020) of the `ZeRO-3` configuration to support efficient large-scale distributed optimization. Computation uses `bfloat16` precision. We integrate `FlashAttention V2` (Dao, 2024) to increase memory efficiency and accelerate attention computation. Each GPU processes a batch size of 4. Gradient accumulation over 8 steps yields a total effective batch size of 256 across 8 GPUs. The initial learning rate is $1 \times 10^{-6}$, which is substantially lower than the learning rates commonly used for similar architectures. The first 10% of training steps are used for warm-up. In this setting, we train the model for 12 epochs to achieve convergence.

**Stage II: Reinforcement Learning** Following the SFT stage, we further optimize the model via reinforcement learning to enable a "think–act–rethink–answer" reasoning procedure. We employ the `verl` framework with the GRPO (Shao et al., 2024) advantage estimator. The actor is trained with a learning rate of $1 \times 10^{-6}$. GRPO optimization uses a mini-batch size of 128. The maximum prompt length is 8,192 tokens. The generated response length is 10,240 tokens. Each episode contains two dialogue turns. We use the `vLLM` (Kwon et al., 2023) inference to generate rollouts. 8 rollout workers run with a GPU memory utilization target of 80%. The maximum batched token capacity per rollout step is 32,768 tokens. To reduce memory usage, the reference model's parameters are fully offloaded to CPU. The actor uses fully sharded data parallel (FSDP) (Zhao et al., 2023) training with both parameter and optimizer state offloading. Training runs for two epochs.

## A.4 CONSTRUCTED TRAINING DATA

Figure 9a depicts a word cloud visualization of the proposed VQA training data, where font size encodes term frequency across the corpus. The analysis reveals two dominant semantic categories. First, spatial descriptors (e.g., coordinate, box, boxes) appear with high frequency, indicating that a substantial subset of questions requires the localization or annotation of anatomical structures, often via bounding-box specifications. Second, clinical diagnostic terms (e.g., tumor, nodules, benign, malignant, along with organ-specific entities) are prominently represented, reflecting the dataset's emphasis on disease-related reasoning.

Figure 9b presents a two-ring diagram summarizing the distribution of medical imaging modalities and their most common diagnostic subcategories in the dataset. The inner ring aggregates the data by modality. MRI accounts for 45% of all studies, followed by ultrasound (15%), CT (12%), X-ray (11%), optical coherence tomography (OCT) (8%), pathology (6%), and electrocardiography (ECG) (3%).The outer ring provides a proportional breakdown of the primary subcategories within each modality. For MRI, kidney tumors constitute 25% of its cases, lung tumors 22%, and normal kidneys 17%, with the inclusion of lung tumors arising from specific MRI protocols at contributing institutions. Ultrasound cases comprise malignant (65%) and benign (35%) tumors, whereas CT data are exclusively nodule cases, predominantly pulmonary. The dataset's coverage spans multiple anatomical regions and disease types, offering substantial diversity. Examples of our constructed training data for SFT are presented in Figures 10 and 11. Examples of our constructed training data for RL are presented in Figures 12 and 13.

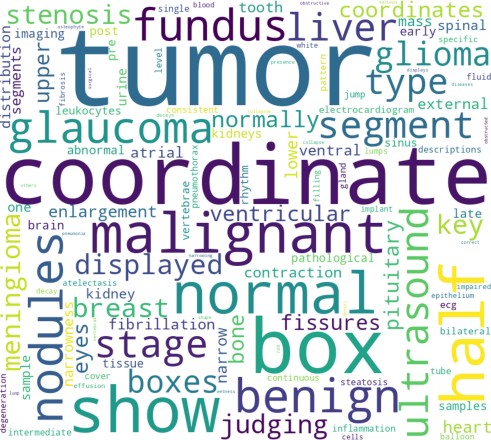
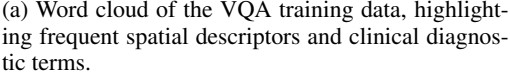

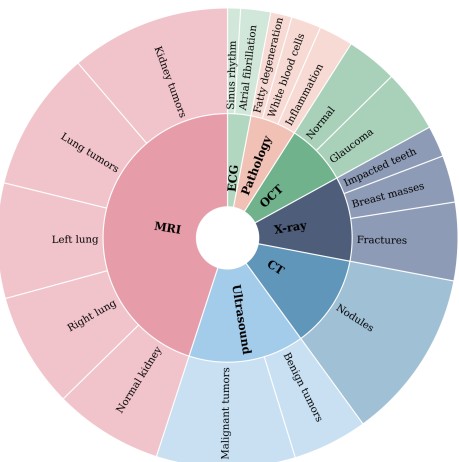

(a) Word cloud of the VQA training data, highlighting frequent spatial descriptors and clinical diagnostic terms.

(b) Distribution of medical imaging modalities and their subcategories in the dataset.

Figure 9: Distribution of our constructed data. Our constructed dataset data on anatomical localization and pathology-oriented reasoning across a wide range of medical imaging modalities and disease types.

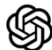 Are there nodules in the right half of the image?
A) Yes B) No

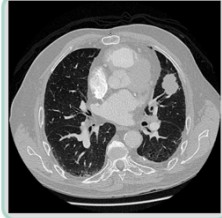

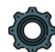 {"**thought**": "In the right half of the image, I observed a structure of a suspected nodule. In order to further confirm its characteristics and accurate location, I will use the marking tool to mark the nodule for detailed analysis.",
"**actions**": [{"name": "Mark", "arguments": {"box": [[462, 218, 518, 285]]}}]}

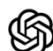 I have marked the region of interest on the image. Please answer the question based on the marked region.

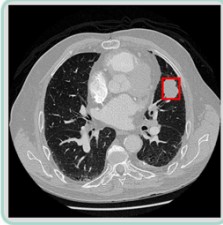

{"**thought**": "Reconfirm the marked nodule area and found that the nodule is clearly visible, with regular morphology, located in the right half of the image, which is consistent with the typical manifestation of the nodule. Based on the annotation and observation, nodules were diagnosed in the right half of the image.",
"**actions**": [{"name": "Terminate", "arguments": {"answer": "A"}}]}

Figure 10: Examples of our constructed data for SFT.

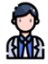 Is there a tumor in the image?
A) Yes B) No

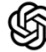

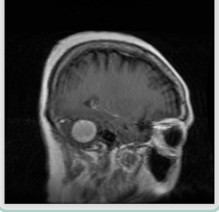

{"**thought**": "A suspicious tumor area was observed in the image, located approximately in the coordinate box [204, 384, 282, 453]. In order to further confirm the characteristics and properties of the tumor, the marking tool needs to be used to mark the area.",
"**actions**": [{"name": "Mark", "arguments": {"box": [[204, 384, 282, 453]]}}]}

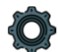 I have marked the region of interest on the image. Please answer the question based on the marked region.

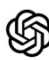

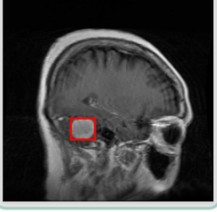

{"**thought**": "Reconfirm the marked tumor area and found that the area is irregular in shape and clear in boundaries, which conforms to the typical image manifestations of the tumor. Combined with the image characteristics, it can be judged that there is indeed a tumor in the image. The final decision is option A.",
"**actions**": [{"name": "Terminate", "arguments": {"answer": "A"}}]}

Figure 11: Examples of our constructed data for SFT.

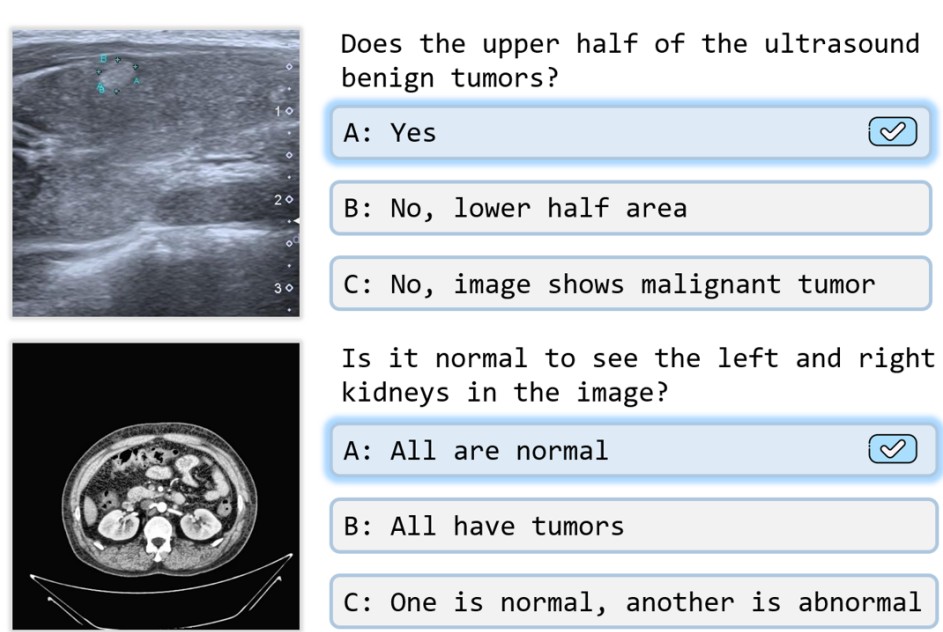

Figure 12: Examples of our constructed data for RL.

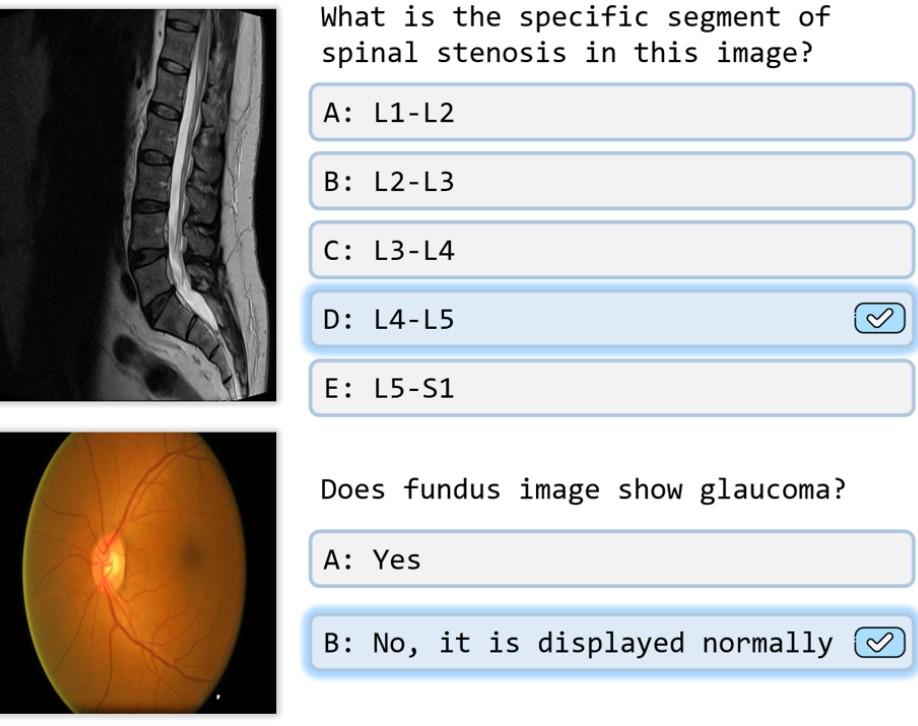

Figure 13: Examples of our constructed data for RL.

## A.5 CASE STUDY

Figure 14 presents an example that illustrates the core workflow of ViTAR. The model integrates structured reasoning with explicit visual grounding through a "*think–act–rethink–answer*" process. In the "think" stage, ViTAR receives a query about the presence of abnormalities in a chest radiograph. It performs an initial inspection of the entire image to gather coarse diagnostic cues. In the act stage, the model visually marks a region of interest in the left lower lung field. The "rethink" stage is guided by this explicit visual cue. ViTAR refines its reasoning and identifies the localized opacity as suggestive of pleural effusion. In the answer stage, the system returns a positive diagnosis, indicating that an abnormality is present. This case demonstrates how ViTAR integrates step-wise reasoning with interactive visual localization. Such coupling produces decisions that are both accurate and interpretable, which is essential for medical human-AI alignment.

In Figure 15, ViTAR analyzes the image to determine whether abnormalities are present. It marks a region of interest in the lower left portion of the image. Based on this highlighted area, ViTAR identifies imaging features consistent with a mass, which constitutes an abnormal finding, and concludes with the answer.

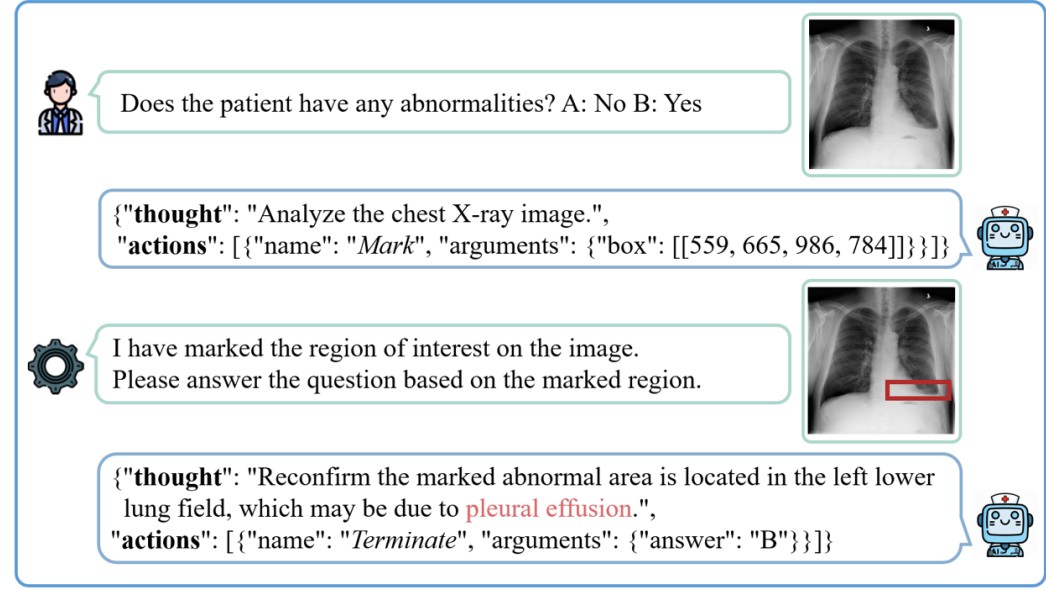

Figure 14: Illustrated example of ViTAR's "think–act–rethink–answer" pipeline in a chest radiograph question answering.

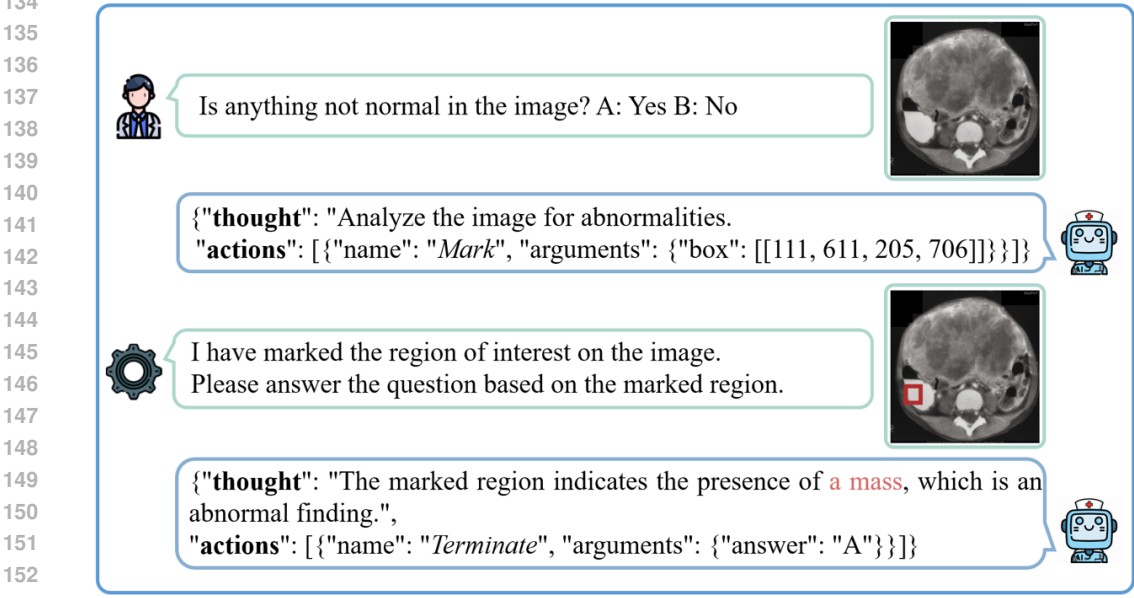

Figure 15: Example of ViTAR identifying a suspected mass within the marked region of interest, leading to an abnormality-positive decision.

### A.6 PROMPTS

The ViTAR model adopts the think-and-act inference paradigm as its default reasoning strategy, with its prompt design shown in Figure 19. In contrast, the direct-answering paradigm omits multi-turn reasoning and outputs the answer in a single step. Its prompt design is presented in Figure 21. For VQA generation, we construct a set of prompts with diverse templates, as illustrated in Figure 17, and design a dedicated verification prompt for validating VQA outputs (Figure 18). We employ GPT-4o (Hurst et al., 2024) to generate intermediate thoughts of VQA, as specified in the detailed prompt shown in Figure 16.

### A.7 THE USE OF LARGE LANGUAGE MODELS

Large language models (LLMs) were used to aid the writing of this work. LLMs helped improve clarity and grammar of the manuscript. LLMs were not used for literature retrieval, related work discovery, or research ideation.

**Prompt for generating thought and re-thought**

You are a medical AI assistant with visual understanding capabilities. When you receive a medical image and a related question, you will first analyze the image content, detecting key anatomical structures or abnormalities (such as fractures, masses, cells, etc.). If abnormalities relevant to the question are detected, describe what you observe and use the mark tool to annotate them; if no targets are found, explain why.

Please generate a structured training sample based on the following input, in JSON format as shown below:

{ "thought 1": "List initially identified abnormal features, specify the regions to be annotated with mark bbox and the reasons, and propose preliminary diagnostic hypotheses that require verification.",

"thought 2": "Re-examine the annotated regions in the image based on the marked areas, provide a detailed diagnostic analysis, and make a final decision." }

Requirements:

1. The thought 1 field should describe what you observe in the image (e.g., fractures, masses, soft tissue swelling, etc. No need for overly detailed descriptions at this stage, as certainty is still low). Explain the next action to be taken, such as marking these areas in the image for further confirmation.

2. The thought 2 field should reconfirm the annotated regions, provide a diagnostic analysis, and finally state the definitive decision.

Figure 16: Prompt for generating thought and re-thought.

**Prompt for generating VQA**

I am constructing a visual multiple-choice question-and-answer set related to chest X-rays. I will provide chest X-ray images with a resolution of 1024*1024, along with the coordinates and categories of atelectasis, cardiomegaly, effusion, infiltration, mass, nodule, pneumonia, and pneumothorax. The question templates are as follows:

Question 1: Does the image show effusion?
Options: A) Yes B) No

Question 2: Does the image show pneumothorax?
Options: A) Yes B) No

Question 3: Does the right half of the image show cardiomegaly?
Options: A) Yes B) No, the left half shows cardiomegaly C) No, the image does not show cardiomegaly (Note: The answer is determined based on the bounding box and image resolution. If it is in the middle, discard this question.)

Question 4: Does the left half of the image show nodules?
Options: A) Yes B) No, the right half shows nodules C) No, the image does not show nodules (Note: The answer is determined based on the bounding box and image resolution. If it is in the middle, discard this question.)

Question 5: The image shows a mass. What are its bounding box coordinates?
Options: A) [213, 175, 239, 211] B) [21, 22, 46, 56] C) [126, 159, 252, 223]
(Note: If there is no mass, discard this question. Modify the bounding box coordinates based on the provided information.)

Question 6: The image shows pneumonia. What are its bounding box coordinates?
Options: A) [213, 175, 239, 211] B) [21, 22, 46, 56] C) [126, 159, 252, 223] (Note: If there is no pneumonia, discard this question. Modify the bounding box coordinates based on the provided information.)

Question 7: What does the coordinate (X, Y) in the image show?
Options: A) Atelectasis B) Cardiomegaly C) Effusion D) Mass E) Other
(Note: (X, Y) should be filled with the correct coordinates based on the provided bounding box information.)

Note: The templates should be modified according to the specific image content. You need to provide the question, options, and answer, and return them in JSON format:

```
[{"Question": "xxx", "Options": "xxx", "Answer": "A/B/C/D/E"}]
```

Figure 17: Prompt for generating VQA.

---

**Prompt for verifying VQA**

You are a medical imaging question-answer (QA) pair validation expert. Your task is to verify whether the given answer is correct based on the provided image information, question, options, and the specified answer. This is a single-choice question, and the answer must be a single option (A/B/C/D/E).

Please strictly evaluate based on the following information:

      1. Image Information: `{image information}`

      2. Image Resolution: `{resolution}`

      3. Question: `{question}`

      4. Options: `{options}`

      5. Answer to Verify: `{answer}`

Carefully analyze the question and options, and use the image information and resolution for reasoning.
Your output must be in one of the following formats:
If the answer format is incorrect (not a single A/B/C/D/E), respond with:
`"Format error: The answer must be a single option A/B/C/D/E."`
If the answer format is correct and the answer is correct, respond only with:
`"Correct."`
If the answer format is correct but the answer is wrong, respond with:
`"Incorrect"`, followed by a brief explanation.

Figure 18: Prompt for verifying VQA.

---

**System prompt for ViTAR**

System prompt: You are a medical image analysis assistant capable of analyzing medical images and answering questions about them. Your goal is to answer questions about medical images including modality, body part, and other medical details. You can rely on your own capabilities or use marking tools to assist in solving. Your output should be in a strict JSON format as follows:
{ "thought": "the reasoning process",
  `"actions":["name":"action","arguments":"argument1":"value1"] }`

Figure 19: System prompt for ViTAR.

---

**User prompt for inference**

Output the thinking process in `<think></think>` and final answer in `<answer></answer>` tags. The output answer format should be as follows:
`<think>` reasoning process here `</think>` `<answer>` answer here (just the letter corresponding to the option, do not provide any explanation) `</answer>`.
Please strictly follow the format.

Figure 20: User prompt for inference.

**User prompt for inference directly**

Output the final answer in `<answer> </answer>` tags. The output answer format should be as follows:
`<answer>` answer here (just the letter corresponding to the option, do not provide any explanation) `</answer>`.
Please strictly follow the format.

Figure 21: User prompt for inference directly.

