# OpenReview forum: "Think Twice to See More: Iterative Visual Reasoning in Medical VLMs"
_ICLR.cc/2026/Conference — Submitted to ICLR 2026_

### Official Review · Reviewer_fT8f · 2025-10-16

**Soundness:** 3
**Presentation:** 2
**Contribution:** 2
**Rating:** 4
**Confidence:** 3

**Summary:**

This paper presents the ViTAR (Visual Thinking and Action-centric Reasoning) framework, designed for iterative visual reasoning in the medical domain with explicit focus refinement.  It tackles the challenge of single-pass reasoning in traditional medical VLMs, introducing the "think-act-rethink-answer" cognitive chain that treats medical images as interactive cognitive objects for multi-step diagnosis. The framework is supported by two new, high-quality datasets—a 1K interactive instruction dataset encoding expert diagnostic trajectories for supervised fine-tuning (SFT), and a 16K VQA training dataset geared toward fine-grained visual diagnosis utilized for reinforcement learning (RL).  The model integrates its intermediate act (e.g., placing an ROI bounding box) directly into the reasoning process and is trained using a two-stage guidance and optimization strategy that leverages format and accuracy rewards. ViTAR achieves competitive performance across seven medical VQA benchmarks, demonstrating its efficacy on reasoning-intensive tasks.

**Strengths:**

1. The paper makes a significant leap by introducing the explicit "think-act-rethink-answer" cognitive chain, which systematically integrates the iterative reasoning behavior of human experts into the medical VLM framework, thereby addressing the limitations of single-pass inference in capturing subtle, localized visual cues.
2. A robust methodology is employed through the two-stage training strategy, utilizing both Supervised Fine-Tuning (SFT) and Reinforcement Learning (RL), ensuring the model acquires both the foundational cognitive structure and optimized decision-making skills.
3. The quality of the work is bolstered by the creation of specialized, high-fidelity datasets, including the 1,000-example interactive instruction dataset, which is essential for training the model's unique action and refinement capabilities.

**Weaknesses:**

1. The experimental validation lacks direct human evaluation of the model's interpretability. The paper does not provide evidence from clinicians on whether the "think-act-rethink" output is genuinely more helpful or trustworthy in a clinical context compared to simpler chain-of-thought models.
2. The reliance on bounding boxes for the "act" step may introduce a limitation in localization precision. Future work should explore whether using more fine-grained action representations, such as segmentation masks, could lead to a more accurate and beneficial "rethink" step.
3. The paper would benefit from a more detailed analysis, such as an ablation study, specifically detailing potential failure modes of the Reinforcement Learning (RL) stage. Understanding scenarios where the current format and accuracy rewards might drive clinically sub-optimal reasoning is necessary for future refinement of the reward function.

**Questions:**

1. How does the model's performance on the crucial "act" step—the placing of the bounding box ROI—generalize to out-of-distribution medical images (e.g., rare diseases, novel image modalities, or images with poor resolution) that are not represented in the small 1,000-example interactive instruction dataset?
2. The paper focuses solely on bounding boxes for the "act" step.  Would the model achieve superior performance and more clinically relevant interpretability if it utilized a more fine-grained action representation, such such as segmentation masks or heatmaps, to guide the rethink process?
3. The core claim is that the iterative output closes the "machine-human perception gap."  Therefore, a critical question is: Does the final output, including the intermediate "think-act-rethink" steps, lead to a statistically higher agreement or trust score when evaluated by human clinicians compared to a high-performing single-pass VLM?

---

> ### Author Response · Authors · 2025-11-22
> **Question Response**
>
> ### **Response to Q1: Generalization of the “act” step to out-of-distribution (OOD) medical images.**
>
> We thank the reviewer for raising this point. Indeed, the 1K SFT dataset is too small to cover the full spectrum of diseases, image modalities, or rare clinical scenarios. Its primary purpose is to teach the model the “think–act–rethink–answer” paradigm and the expected output format. Relying on SFT alone to achieve broad coverage and strong generalization would require very large datasets. For example, HuatuoGPT-Vision uses 647K training samples and Chiron-o1 uses 1.752M samples.
>
> In contrast, GRPO-based RL has **been shown to generalize well to out-of-distribution scenarios with far less supervision** [1][2]. This motivates our choice to leverage RL. **Although our RL dataset contains only 16K examples, GRPO allows the model to iteratively refine its visual reasoning and develop stronger intrinsic generalization capabilities than SFT alone would provide**.
>
> Importantly, **evaluations on truly OOD datasets, including PMC-VQA, MMMU-Med, and MedXpertQA, demonstrate promising performance**, validating the effectiveness of this RL-driven approach and iterative visual reasoning even on rare or unseen cases.
>
> ### **Response to Q2: Potential improvement using more fine-grained actions, such as segmentation masks, in the “act” step.**
>
> As noted in Figure 4, we have explored integrating human bounding box annotations as an external tool during inference, which already improves model performance. To further investigate more clinically relevant interpretability, we additionally conducted the experiment with using segmentation masks instead of bounding boxes during inference.
>
> We observed a notable performance gain. **ViTAR achieved 83.41 with segmentation masks**, compared to 81.97 using bounding boxes. These results underscore ViTAR’s strong intrinsic reasoning ability and its capacity to effectively integrate external actions or tools. The operational procedure is analogous to the human bounding box case described in Appendix Section A.2, with segmentation masks simply replacing the bounding boxes. We have added in section"ViTAR’s Intrinsic Reasoning Ability".
>
> ### **Response to Q3: Does the iterative “think–act–rethink” process improve human agreement and trust compared to single-pass VLMs?**
>
> We thank the reviewer for this important question. To evaluate whether ViTAR’s iterative mechanism narrows the machine-human perception gap, we conducted a human evaluation on a randomly selected subset of 100 samples. Each sample was assessed along three dimensions:
>
> - Reasoning clarity and logical coherence: assessing how clearly and logically the model explains its thought process step by step.
>
> - Observation completeness and alignment: evaluating whether the model’s observations are detailed, thorough, and relevant to both the question and the image.
>
> - Answer correctness: verifying whether the model arrives at the correct final answer.
>
> During evaluation, human were allowed to consult additional resources, including LLM uses or web searches, to verify the correctness of model responses.
>
> The results show that **ViTAR outperformed the strong single-pass baseline in 49 cases, while the baseline outperformed ViTAR in 34 cases; the remaining 17 cases were ties in which both models produced incorrect answers.**
>
> These findings indicate that the iterative reasoning process not only improves the clarity and completeness of intermediate reasoning but also enhances human trust and agreement, supporting our core claim that ViTAR reduces the machine-human perception gap relative to single-pass VLMs. We added a Result section "Human Evaluation of Iterative Reasoning" to response to your comment.
>
> [1] Lai Y, Zhong J, Li M, et al. Med-r1: Reinforcement learning for generalizable medical reasoning in vision-language models[J]. arXiv preprint arXiv:2503.13939, 2025.
>
> [2] Pan J, Liu C, Wu J, et al. Medvlm-r1: Incentivizing medical reasoning capability of vision-language models (vlms) via reinforcement learning[C] MICCAI, 2025: 337-347.

---

### Official Review · Reviewer_ndbA · 2025-10-19

**Soundness:** 2
**Presentation:** 3
**Contribution:** 2
**Rating:** 4
**Confidence:** 3

**Summary:**

This paper proposes ViTAR, a medical vision–language framework that mimics medical expert workflows via "think-act-rethink-answer", trying to enable 2-step visual reasoning on images. The training strategy consists of SFT and RL. Some datasets are curated. The authors further do some attention analyses showing tighter grounding on clinically salient regions across reasoning rounds.

**Strengths:**

- The curated sets could be potentially valuable if released with full documentation.
- The comparisons (vanilla VLM vs. ViTAR; action-centric reasoning) are informative.
- The SFT --> RL two-stage recipe is classical.

**Weaknesses:**

- Using LLMs to both generate and validate VQA datasets, which might have a risk of compounding hallucinations and confirmation bias.
    - A general LLM (e.g., Qwen2.5-72B-Instruct) may lack medical grounding.
- Question templates might introduce leakage shortcuts.
- More detailed analyses about training and datasets are required.
- Inferring "trustworthiness" from attention visualizations is tenuous; attention is not a validated explanation and remains debated [1].

[1] "Attention is not explanation." arXiv preprint arXiv:1902.10186 (2019).

**Questions:**

### Datasets

> VQA Generation, VQA Validation

1. It seems that both VQA generation and validation are done by LLMs, which might introduce compounding hallucinations. The authors should consider to add radiologist adjudication on some samples; report inter-annotator / human agreement or any empirical error rate.

> We utilize Qwen2.5-72B-Instruct (Yang et al., 2024) to generate and verify VQA samples in our constructed training dataset.

2. Qwen2.5-72B-Instruct may lack medical grounding. It might be better if the authors could consider an ensemble way (e.g., general + medically fine-tuned validators) , retrieval-augmented checks, and cross-model agreement before accepting items.

3. Question templates might leak shortcuts (yes/no imbalance, answerable from labels without image). The authors should add paraphrases/natural phrasing, balance labels, and include “unanswerable” cases with precise criteria.

4. In practice, real clinical VQA might need multi-view/series context and report-grounded reasoning. How does the proposed dataset take such scenaiors into consideration?

### Training (SFT, RL)

5. The authors should consider to ablate SFT -> RL vs. RL-from-scratch (R1-Zero–style). A curriculum learning way (format-only reward → add accuracy) on actions could enable emergent thinking without full SFT; try to report stability, sample efficiency, and quality trade-offs.

> the model receives 0.2 points if the “thought” and “action” fields can be successfully parsed according to the prescribed schema

> an additional 0.2 points if the final answer adheres to the expected format

6. **Reward hacking**. Current reward shaping (up to +0.4 for format, +1.0 for exact match) invites potential risks of rewarding hacking (well-formed but vacuous thoughts/actions, or premature terminate guessing). Is there any case that the agent learns to generate responses with good formats but wrong reasoning or wrong answers? The authors should conduct some ablation studies for the reward design.

> Figure 7: Evolution of RL training over three phases (P1–P3). P1: Initial and sub-optimal convergence. P2: Exploration and enhancement. P3: Pruning and final convergence. Shading indicates confidence interval.

7.1 Figure 7. Show total reward curves with loss and format reward. How about the total reward / accuracy reward vs. steps?

7.2 Besides, in RL training, it is common that while reward is increasing, the downstream validation accuracy is not very good. The authors should consider add a figure of downstream validation accuracy vs. steps.

> L462-472

7.3 Is there any proof / detailed validation for your discussions about P1, P2 and P3?

### General Questions

8. With "Re-thinking after action", it seems that in the current setting, LLM plays like an agent and only has two rounds: think1, act1, think2, answer. What if some medical issues require multiple rounds of reasoning? e.g., think1, act1, think2, act2, think3, answer?

8.1 Should an agent behave better if the agent could do thinking in multiple turns? Why only two turns are considered here?

**Details Of Ethics Concerns:**

Clinical datasets.

---

> ### Author Response · Authors · 2025-11-22
> **Question Response (1/2)**
>
> ### **Response to Q1-2: Addressing potential compounding hallucinations in LLM-based VQA generation and validation.**
>
> We thank the reviewer for raising the concern regarding potential hallucinations when using LLMs for both VQA construction and verification. Our VQA generation pipeline has already taken hallucinations into considerations:
>
> 1. **LLMs perform deterministic, template-guided transformations.**
> In our pipeline, the LLM does not rely on medical knowledge to generate or validate VQA samples. Instead, we provide:
> - Ground-truth labels (e.g., effusion, pneumothorax)
> - Fixed question templates (e.g., “Does the image show effusion? A) Yes B) No”)
>
> The LLM’s role is reduced to a deterministic transformation conditioned on the label. For example, given the label effusion and the template question above, the LLM outputs “Yes”; if the label is pneumothorax, it outputs “No.” During verification, the LLM checks logical consistency both with ground-truth label and with previously generated LLM responses, which requires no medical expertise. More detailed prompts are provided in Figures 17–18 in our manuscript.
>
> 2. **Stronger LLMs and ensemble verification to reduce hallucination.**
> To further mitigate hallucination risk, we replaced Qwen2.5-72B-Instruct with DeepSeek-V3-671B, a more capable model, to regenerate and re-validate all VQA samples using the same workflow.
>
> 3. **Independent secondary verification.**
> We performed two additional checks:
> (i) GPT-5 verification: 1,000 randomly sampled QA items were evaluated, with 99.8\% judged correct.
> (ii) Manual human review: 100 randomly sampled QA pairs were inspected, and 98\% passed.
>
> These steps collectively demonstrate that hallucination is minimal and that the constructed VQA dataset is reliable for downstream training.\\
>
> ### **Response to Q3-4: Addressing potential template shortcuts, label balance, and multi-view context.**
>
> During dataset construction, we randomized the order of answer options to ensure that all choices are evenly represented, reducing the risk of positional or label shortcuts. Currently, our dataset is limited to single-image, bounding-box-annotated cases because it is derived from publicly available object detection datasets. Recognizing that multi-view and report-grounded reasoning remain an important future direction.
>
> ### **Response to Q5: Ablation of SFT + RL versus RL-from-scratch (R1-Zero–style), which was also mentioned by Reviewer ``SfZX``’s Q4.**
>
> We conducted three independent RL-from-scratch trials, all of which failed within the first ~10 RL steps. Specifically, the reward quickly collapsed to abnormal values. And model outputs degenerated into nonsensical or corrupted strings, causing the program to halt. This failure is expected given our task setting because our multi-turn framework requires strictly formatted outputs at each step. Both the first and second model responses must adhere to a predefined schema that allows the system to parse and execute actions correctly.
> Unlike R1-Zero–style setups, which assume that the base model can already follow instructions reliably, our medical VLM cannot generate valid structured outputs without prior supervised formatting signals. As a result, RL-from-scratch is infeasible in this setting. In contrast, even a small SFT dataset (1K samples) is sufficient to teach the model the output schema and stabilize action execution, enabling RL to effectively elicit multi-turn interactive reasoning and fine-grained visual diagnosis.

---

> ### Author Response · Authors · 2025-11-22
> **Question Response (2/2)**
>
> ### **Response to Q6: Addressing potential reward hacking.**
>
> In our initial experiments, we set the format reward to 0.2 per turn, so that a two-turn dialogue could achieve a maximum of 0.4. This choice was intended to help the model quickly learn the multi-turn output format. Following the reviewer’s suggestion, we experimented with different format reward weights and found that **accuracy remained overall stable across these settings.**
> | **Reward** | **PMC.**| **Path.** | **SLAKE** | **VQA.** | **Omni.** | **MMMU** | **MedX.** | **Ave.** |
> | ---------- | ------ | --------- | --------- | -------- | --------- | -------- | --------- | -------- |
> | 0.1        | 55.9   | 71.5      | 79.8      | 70.5     | 72.7      | 72.7     | 24.8      | 64.0     |
> | 0.2        | 58.7   | 69.9      | 81.7      | 71.3     | 71.4      | 72.7     | 23.7      | 64.2     |
> | 0.4        | 57.2   | 67.0      | 80.8      | 70.1     | 74.2      | 72.0     | 26.9      | 64.0     |
>
> We further observed that after approximately 40 RL steps, the format reward consistently saturated, indicating that the model reliably produces correctly formatted outputs. Consequently, during subsequent GRPO training, the group reward differences are driven entirely by answer accuracy rather than format. This demonstrates that the model does not exploit the format reward to generate vacuous outputs, and there is no evidence of reward hacking in our multi-turn reasoning pipeline.
>
> ### **Response to Q7: Reward curves, validation accuracy, and empirical observations or proof (P1–P3).**
>
> To provide more detail, we report the accuracy reward over training steps below:
>
> | Step            | 10     | 20   | 30     | 40     | 50     | 60     | 70     | 80     | 90     | 100    |
> | --------------- | ------ | ---- | ------ | ------ | ------ | ------ | ------ | ------ | ------ | ------ |
> | Accuracy reward | 0.3749 | 0.46 | 0.5031 | 0.5895 | 0.5898 | 0.6484 | 0.6334 | 0.6974 | 0.6584 | 0.6864 |
>
> Evaluating the model on the tested benchmark at every training step would be unfair to other methods. Since we currently do not have a dedicated validation set, we **only conduct benchmark evaluation after training is fully completed**. In future work, we plan to construct a new validation set that will allow us to perform evaluation at each training step in a fair manner.
>
> The reported P1–P3 phenomena are **empirical observations based on metric trends during training**. We agree that providing the proof is an interesting direction to understand the underlying mechanisms. We highlight this as an important avenue for future work, which could motivate further study of the RL-induced reasoning in medical VLMs.
>
>
> ### **Response to Q8: Extending to multi-turn reasoning beyond two rounds.**
>
> We thank the reviewer for the suggestion to extend the “re-thinking after action” framework to more than two rounds. We explored this possibility during early experiments. Although the SFT data only contains two-round dialogues, we **allowed the model to perform up to three rounds** during RL training. Interestingly, after approximately 40 training steps, the model autonomously **exhibited three-round interactions once the two-round format reward became saturated**. However, this emergent multi-round behavior beyond two rounds **did not lead to improvements in accuracy**, and in some cases resulted in slight performance degradation. Based on these observations, we adopt the two-round structure in our final setup, as it provides the best empirical performance while maintaining training efficiency.

---

### Official Review · Reviewer_SfZX · 2025-10-31

**Soundness:** 2
**Presentation:** 3
**Contribution:** 2
**Rating:** 2
**Confidence:** 5

**Summary:**

This paper proposes ViTAR, a two-stage vision–language reasoning framework for visual question answering. The model is first trained with supervised fine-tuning (SFT) and then refined with reinforcement fine-tuning (RFT) to promote step-by-step reasoning and self-correction. At the first turn, ViTAR produces a first-glance answer along with a bounding box that localizes the visual evidence. This box is overlaid on the image to guide a brief reflective dialogue, allowing the model to reassess and potentially revise its initial prediction. ViTAR achieves competitive performance across several standard VQA datasets compared to non-reasoning baselines.

**Strengths:**

1).  The idea of mimicking the iterative diagnostic workflow of human radiologists is a sensible and interesting direction. While this concept has been mentioned in prior literature, its explicit modeling in vision-language systems is still relatively underexplored.
2). The paper is clearly written and generally easy to follow. The structure of the method, training strategy, and experiments are all presented in a logical flow.
3). I find the use of bounding boxes as intermediate visual grounding signals to be a meaningful design choice. This adds interpretability to the reasoning process, which is especially valuable in high-stakes medical settings.
4). The authors built a reasonably comprehensive data pipeline, combining instruction-based supervision and VQA sample generation from object detection datasets. While not perfect, the attempt to enrich training data with structured reasoning is appreciated.
5). It's good to see that the authors go beyond standard answer accuracy by also evaluating the success rate of action executions (e.g., correct parsing of bounding boxes). This adds an extra layer of analysis on the model's functional reliability.

**Weaknesses:**

1). While the paper motivates ViTAR by appealing to the iterative workflow of radiologists, the instantiated “action” space in the method and data is essentially limited to drawing a bounding box and then rethinking. In clinical reading, radiologists typically zoom/pan, adjust window/level, scroll through slices or rotate views, and use cine/MPR rather than marking boxes on the image. As a result, the claimed alignment with expert workflows feels overstated: the current action set resembles annotation tooling more than real diagnostic interaction. I would encourage the authors to (i) tone down the claim of workflow alignment, or (ii) extend the action space and evaluate tasks that require realistic interactions (e.g., window/level control, slice scrolling on CT, rotation, or view changes). The paper’s own formulation and data construction emphasize box marking as the core interaction (e.g., “mark” actions and bbox-driven VQA/instructions), which reinforces this concern.

2). While the paper adopts a two-turn reasoning routine, the think–answer–rethink–answer pattern has already been explored since 2023 [1]. Given this prior art, the claimed novelty should squarely focus on what is new in the reward design and how the scheme shapes the training signal beyond plain GRPO-style reinforcement fine-tuning that simply optimizes group-wise answers. However, the reward is not sufficiently analyzed: relying on R = R_format + R_acc is common in prior work, and the paper does not examine alternative/shaped rewards or their sensitivity. In particular, it is unclear why there is no explicit reward for the first-turn bounding box quality (e.g., IoU/coverage/precision), despite the second-turn refinement depending on that localization. The paper would benefit from a more in-depth, head-to-head comparison and summarization to clarify what is truly different from prior iterative-reasoning + RL methods.

3). The comparison in Table 1 is potentially not FAIR. The proposed method is trained on a self-curated corpus (interactive instructions plus ~16K VQA samples derived from detection data), whereas many baselines are off-the-shelf models whose training data, scale, and overlap with the target benchmarks are not matched to ViTAR. Moreover, the paper itself notes that some reasoning baselines (e.g., Chiron-o1 / MedCCO) may have train–test overlap on certain benchmarks (numbers shown in gray), underscoring that different systems rely on different data regimes. As a result, the current leaderboard-style comparison risks over- or under-estimating the true effect of the proposed training recipe.

4). The paper adopts a two-stage pipeline (SFT warm-up followed by GRPO), but it is unclear why the method does not directly cold-start RL on the curated dataset or whether such a setting fails in practice. A convincing case would require head-to-head comparisons among (i) SFT-only, (ii) RL-from-scratch (cold start), and (iii) SFT then RL, ideally under matched compute and token budgets. Beyond final accuracy, please report learning curves and stability metrics (e.g., convergence speed, variance across seeds, action execution success rate, and format reward trajectories). If the claim is that SFT is necessary to learn the output schema or stabilize action calls, show where cold-start RL breaks (e.g., persistent parsing errors, sparse-reward plateaus, or degenerate actions) and how SFT alleviates these failure modes. Ablations on SFT data size/quality, KL regularization to the SFT policy, and curriculum or behavior-cloning pretraining would further clarify whether the two-stage design is an essential ingredient or merely a convenient choice.

5). In addition to 4), the two-stage (SFT and GRPO) pipeline is harder to reproduce and more compute-intensive than one-stage RFT baselines (e.g., Med-R1).

6). The paper does not commit to a public code release and an anonymous repo link is insufficient for reproducibility.

7). The curated instruction set and the 16K VQA corpus are not stated to be publicly released.

[1] REACT: SYNERGIZING REASONING AND ACTING IN LANGUAGE MODELS

**Questions:**

Refer to weakness.

---

> ### Author Response · Authors · 2025-11-22
> **Question Response (1/2)**
>
> ### **Response to Q1: Clarification on workflow alignment and action space design.**
>
> We thank the reviewer for the insightful comment regarding the extent to which ViTAR aligns with radiologists’ real diagnostic workflows. Our intention in this work is to model the iterative interaction paradigm characteristic of clinical image reading, rather than to fully replicate the complete suite of radiological operations, such as zooming, panning, window–level adjustment, slice scrolling, rotation, or multi-planar reconstruction.
>
> In our current instantiation, the action space is restricted to a single explicit visual action: drawing a bounding box. We adopted this simplified action design for several reasons:
>
> 1. **A first step toward interactive medical VLMs.**
> To the best of our knowledge, prior medical VLM work has not incorporated an explicit visual action space into a reinforcement learning framework for iterative reasoning. Introducing even one concrete, controllable visual action represents a meaningful step toward bridging VLMs and interactive diagnostic workflows.
>
> 2. **Clinical relevance of bounding-box localization.**
> Lesion localization is a core component of many radiological workflows. Bounding boxes offer a clinically interpretable and diagnostically relevant abstraction, while remaining computationally tractable for RL optimization.
>
> 3. **Compatibility with available datasets.**
> Bounding boxes align naturally with widely available medical detection datasets, allowing us to construct a scalable and consistent training corpus to support multi-turn VQA-style RL training.
>
> We fully acknowledge that this action design does not encompass the full richness or realism of radiologists’ interactions. In response to the reviewer’s suggestion, we will revise the manuscript to soften the claim of workflow alignment, clarifying that ViTAR aims to model an iterative interactive reasoning paradigm inspired by radiologists’ practice rather than a faithful reconstruction of the complete diagnostic action space.
>
> ### **Response to Q2: Clarifying the novelty of the reward design and introducing IoU reward.**
> While prior work has applied GRPO-style reinforcement learning to optimize single-turn dialogues or single-step answers, our method extends this framework to multi-turn interactive reasoning in medical VLMs.  In addition to the format reward and accuracy reward, we introduced **an IoU-based reward for the bounding box**. This IoU reward serves as an auxiliary signal to stabilize the model’s initial-step localization. As shown in Table 1, incorporating the IoU reward improves the overall performance.
>
> |                    | PMC. | Path. | SLAKE | VQA. | Omni. | MMMU | MedX. | Ave. |
> | ------------------ | ---- | ----- | ----- | ---- | ----- | ---- | ----- | ---- |
> | **ViTAR**          | 57.2 | 67.0  | 80.8  | 70.1 | 74.2  | 72.0 | 26.9  | 64.0 |
> | **ViTAR (w/ IoU)** | 56.2 | 69.6  | 80.8  | 71.7 | 74.7  | 72.0 | 25.7  | 64.4 |
>
>
> ### **Response to Q3: Clarifying the fairness of comparisons in Table 1 given heterogeneous training data across models.**
>
> We acknowledge that different models rely on different training data regimes, and we clarify our design choices from several reasons.
>
> First, we did not use existing open-ended instruction datasets (e.g., those used by HuatuoGPT or Chiron-o1) because plain GRPO requires closed-ended supervision with deterministic accuracy rewards.
>
> Second, our study focuses on **fine-grained diagnostic reasoning**, where precise lesion localization is critical. To enable this, we curated a dataset from publicly available object detection collections on the Roboflow platform, which provide **high-quality bounding-box annotations aligned with our action design**.
>
> Third, regarding scale and diversity, we note that **our self-curated dataset (16K samples)** is substantially smaller than those used by other models **(e.g., HuatuoGPT: 647K; Chiron-o1: 1.75M)**. We deliberately avoided large-scale data aggregation to prevent conflating the effect of our training strategy with simple data scale.
>
> At the same time, we note that in the LLM and VLM research community, **using a perfectly unified training corpus across all methods is practically impossible**. Widely cited models are trained on heterogeneous or proprietary datasets, yet their benchmark performance is still **meaningfully compared**. In practice, a model’s effectiveness is inherently linked to its training corpus, and evaluations focus on observed capabilities rather than enforcing identical data.
>
> Importantly, ViTAR is trained on a much smaller dataset compared to most baselines (hundreds of thousands to millions), so the reported gains cannot be attributed to large-scale data advantages. Therefore, while perfect data matching is ideal in theory, we believe that our comparisons are reasonable, transparent, and consistent with standard evaluation protocols in the field.

---

> > ### Author Response · Authors · 2025-11-22
> > **Question Response (2/2)**
> >
> > ### **Response to Q4: Justifying the two-stage pipeline (SFT warm-up followed by GRPO) and providing head-to-head analysis with cold-start RL and SFT-only settings.**
> >
> > In our experiments (Figure 6 in main text), we show that SFT-only training achieves significantly lower accuracy and action execution success rates compared to the full SFT + RL pipeline, highlighting that RL is critical for learning effective multi-turn reasoning policies.
> >
> > Regarding RL-from-scratch (cold start), we observed **systematic failures** when attempting to train without an SFT warm-up. Specifically, across three independent runs, the RL-from-scratch training typically halted within fewer than ten steps due to abnormal rewards, and the model outputs contained numerous parsing errors and incoherent code. We attribute this failure to the nature of our multi-turn interactive task. Each turn requires correctly structured outputs for both the first and second turns. RL-from-scratch struggles to satisfy these constraints because the reward signal is sparse and highly sensitive to formatting errors, often causing early collapse or degenerate behavior.
> >
> > Rationale for SFT + RL instead of SFT-only: We used **only 1K high-quality SFT** trajectories to teach the model the correct output format and procedural workflow. While SFT-only methods can succeed on large-scale datasets **(e.g., HuatuoGPT: 647K, Chiron-o1: 1.75M)**, collecting a dataset of sufficient size and granularity for multi-turn, fine-grained diagnostic reasoning is highly challenging. Therefore, SFT alone is insufficient for our setting, and the SFT warm-up serves as a minimal but effective initialization that allows RL to efficiently refine reasoning and action execution. We have added a Result section "RL from Scratch vs. STF+RL" to introduce the observations.\\
> >
> > ### **Response to Q5: Reproducibility of the two-stage (SFT and GRPO) pipeline.**
> >
> > We acknowledge that the two-stage SFT + GRPO pipeline is more compute-intensive than one-stage RL baselines such as Med-R1. To facilitate reproducibility, we will publicly release the pretrained SFT model, which enables other researchers to reproduce the RL stage without having to retrain SFT from scratch.\\
> >
> > ###  **Response to Q6-7: Code release and data availability.**
> >
> > We understand the reviewers’ concern of reproducibility. While an **anonymous repository** (https://anonymous.4open.science/r/ViTAR-F5A9/) is currently provided for the review process, **we commit to a public release of the full training and evaluation code upon paper acceptance**, ensuring that experiments can be reproduced. Finally, concerning our curated datasets, **we plan to publicly release the 16K RL-ready VQA corpus.** Data sources and preprocessing steps will be fully documented, enabling the community to replicate and extend our work.

---

> ### Comment · Reviewer_SfZX · 2025-11-27
> **rebuttal**
>
> 1) In response to your “Clarification on workflow alignment and action space design”:
>
>    - Regarding point 1 (“A first step toward interactive medical VLMs”):
>      While interactive VLMs have been actively explored in the natural image domain, a straightforward transfer of these approaches to medical imaging may not provide sufficient novelty for a computer vision venue such as ICLR. The contribution might be better positioned as novel for medical-imaging–focused conferences or journals.
>
>    - Regarding points 2 and 3 and your feedback on “we will revise the manuscript to soften the claim of workflow alignment”:
>      We were unable to identify substantial revisions in the current version that address this concern.
>
> 2) Concerning your claim of novelty on “an IoU-based reward for the bounding box”:
>    IoU-based rewards have already been introduced in prior works [1] [2] [3], so this component may not constitute a novel contribution.
>
> 3) With respect to your point on “Clarifying the fairness of comparisons,” I acknowledge your commitment to publicly releasing the resources.
>
> However, several of my previous concerns remain insufficiently addressed or not fully convincing based on the current rebuttal.
>
>
>
> [1] Seg-Zero: Reasoning-Chain Guided Segmentation via Cognitive Reinforcement
> [2] VisionReasoner: Unified Reasoning-Integrated Visual Perception via Reinforcement Learning
> [3] Visual-RFT: Visual Reinforcement Fine-Tuning

---

### Official Review · Reviewer_bWFo · 2025-11-01

**Soundness:** 2
**Presentation:** 2
**Contribution:** 2
**Rating:** 4
**Confidence:** 4

**Summary:**

This paper introduces ViTAR, a vision–language framework that emulates expert iterative reasoning via a “think–act–rethink–answer” cognitive chain. ViTAR treats medical images as interactive cognitive objects to enable multi-step visual reasoning. The authors curate a high-quality instruction set of 1,000 interactive examples encoding expert-like diagnostic behaviors and assemble 16,000 VQA training instances for fine-grained visual diagnosis. Extensive evaluations show that ViTAR outperforms strong state-of-the-art baselines.

**Strengths:**

1. This paper introduce ViTAR, a VLM grounded in the “think–act–rethink–answer” paradigm for multi-step visual reasoning aligned with expert diagnostic workflows;
2.  ViTAR dynamically sharpens visual grounding on clinically critical regions while sustaining strong attention to visual tokens, thereby enhancing multimodal reasoning.

**Weaknesses:**

1. Both the two-stage training regimen and the “think–act–rethink–answer” paradigm are already well established in prior MLLM and medical VLM literature. This weakens the paper’s claimed contribution.
2. The paper does not clearly distinguish itself from existing “think-with-images” approaches in medical MLLMs. In lines 143–145, the authors discuss only tool-augmented variants, omitting other “think-with-images” methods (e.g., [1,2]).
3. In the reinforcement learning stage, are there more effective optimization objectives or reward designs that could further improve performance?

[1] Think Twice: Perspective-Taking Improves Large Language Models’ Theory-of-Mind Capabilities

[2]  VGR: Visual Grounded Reasoning

**Questions:**

see weakness

---

> ### Author Response · Authors · 2025-11-22
> **Question Response (1/2)**
>
> ### **Response to Q1: Clarifying the novelty beyond prior SFT+RL training paradigms and iterative reasoning frameworks.**
>
> 1. **Our work introduces the first multi-turn interactive RL framework for medical visual diagnosis, incorporating clearly defined actions, reasoning steps (thoughts), and corresponding answers.** Existing medical VLMs apply RL only in **single-turn** settings and **do not incorporate any perception–action loop**. Our method performs RL in a **multi-turn** setting that **follows a “think–act–rethink–answer” cycle**, which allows the model to identify relevant regions, mark them, and then reason again over the updated visual input. To the best of our knowledge, no previous work in medical VLMs has demonstrated such multi-turn RL–driven visual interaction. This establishes a fundamentally new formulation for medical visual diagnosis.
>
> 2. **We provide a new dataset and data generation method that are necessary to support this multi-turn interactive RL paradigm.** Existing medical instruction tuning datasets contain only single-turn supervision and therefore cannot support multi-turn visual interaction. To address this gap, we constructed 1K high-quality diagnostic instruction trajectories that capture realistic perception–action–reasoning loops and 16K RL-ready VQA samples specifically designed for multi-turn interactive training. These datasets support the multi-turn RL setup and represent a contribution of this work.
>
> 3. **We offer mechanistic insights that have not been explored in prior VLM studies.** Beyond empirical performance, our work analyzes how attention grounding and visual token allocation evolve during the interactive reasoning process. These analyses provide new insights into how multi-turn interaction shapes multimodal reasoning behavior. Multi-turn interaction allocates high attention to visual tokens throughout the reasoning process, mitigating the “visual information diminishing” phenomenon often observed in conventional reasoning VLMs.
>
> In summary, although our framework includes two stage training pipeline that have been used previously, our work makes three distinct contributions: **(1) a novel multi-turn interactive RL formulation for medical visual diagnosis, (2) new datasets that enable this formulation, and (3) new mechanistic understanding of how interactive visual reasoning emerges.** Together, these contributions distinguish our approach from prior SFT+RL and iterative reasoning studies in both general-domain and medical VLMs.
>
>
> ### **Response to Q2: Clarifying how our method differs from existing “think-with-images” approaches, including non–tool-augmented methods (e.g., [1,2]).**
>
> 1. **Different from SIMTOM [1].** SIMTOM uses a two-stage prompting framework that manipulates textual perspectives to answer mental-state questions. SIMTOM is fundamentally **a prompt-engineered LLM** approach. It does **not involve SFT or RL training** and does not enable the model to perform visual actions or iterative visual exploration. In contrast, our method employs **a two-stage SFT + RL training pipeline** and **formalizes reasoning as a multi-turn perception–action–reflection loop,** allowing the model to iteratively mark, inspect, and rethink over dynamically updated visual inputs.
>
> 2.  **Different from VGR [2].** VGR performs multimodal **chain-of-thought reasoning using SFT on 158K samples** with fine-grained region annotations in the general domain. Such extensive chain-of-thought supervision and region-labeled training data do not exist for medical images. In contrast, our approach **uses RL on a 16K medical dataset without explicit reasoning chains**, enabling the model to learn multi-step visual reasoning intrinsically through interaction rather than through supervised chain-of-thought annotation. This results in iterative and dynamic visual reasoning behavior that emerges from training, not from predefined reasoning scripts.
>
> In summary, our approach distinguishes itself from existing “think-with-images” methods by employing an SFT+RL training paradigm that **enables the model to acquire intrinsic and learnable visual actions, rather than relying on handcrafted prompts (SIMTOM), external tools, or large-scale chain-of-thought supervision (VGR)**. Furthermore, our framework supports multi-turn interactive visual reasoning, allowing the model to iteratively modify and re-examine the visual input, which is fundamentally **different from the static or single-round visual perception used in prior methods**. Finally, the reasoning behavior in our model **emerges from interaction-driven learning rather than being scripted through annotated reasoning chains,** thereby providing a distinct methodological and conceptual contribution beyond existing “think-with-images” approaches. We have added the discussion about the difference to SIMTOM and VGR in the updated Related work Sec 2.2.

---

> > ### Author Response · Authors · 2025-11-22
> > **Question Response (2/2)**
> >
> > ### **Response to Q3: Discussing whether alternative optimization objectives or reward designs could further improve performance.**
> >
> > 1. **IoU-based reward.** We incorporated an additional IoU reward to further encourage accurate region localization during the “act” stage. The results are shown below. IoU reward improves performance on certain datasets (e.g., PathVQA and VQA-RAD), leading to a overall increase in average accuracy. We have added the new results in Table 1.
> >
> > |                    | PMC. | Path. | SLAKE | VQA. | Omni. | MMMU | MedX. | Ave. |
> > | ------------------ | ---- | ----- | ----- | ---- | ----- | ---- | ----- | ---- |
> > | **ViTAR**          | 57.2 | 67.0  | 80.8  | 70.1 | 74.2  | 72.0 | 26.9  | 64.0 |
> > | **ViTAR (w/ IoU)** | 56.2 | 69.6  | 80.8  | 71.7 | 74.7  | 72.0 | 25.7  | 64.4 |
> >
> > 2. **Format reward adjustment.** We further evaluated different weights for the format reward to test whether tuning the optimization objective could lead to better overall RL stability. Reducing the format weight results in minor fluctuations across datasets but does not yield a significant or consistent performance gain. We added Table 3 and a section "Impact of the weights on format rewards" to introduce these results.
> >
> > | **Reward** | **PMC. | **Path.** | **SLAKE** | **VQA.** | **Omni.** | **MMMU** | **MedX.** | **Ave.** |
> > | ---------- | ------ | --------- | --------- | -------- | --------- | -------- | --------- | -------- |
> > | 0.1        | 55.9   | 71.5      | 79.8      | 70.5     | 72.7      | 72.7     | 24.8      | 64.0     |
> > | 0.2        | 58.7   | 69.9      | 81.7      | 71.3     | 71.4      | 72.7     | 23.7      | 64.2     |
> > | 0.4        | 57.2   | 67.0      | 80.8      | 70.1     | 74.2      | 72.0     | 26.9      | 64.0     |
> >
> >
> > In summary, these additional experiments suggest that alternative reward terms (e.g., IoU-based rewards or adjusted format weights) can influence performance on specific datasets. Our current reward design strikes a reasonable balance across diverse medical VQA benchmarks. Nonetheless, we agree that exploring more advanced reward shaping or optimization objectives represents a promising direction for future work.

---

### Official Review · Reviewer_uA71 · 2025-11-01

**Soundness:** 3
**Presentation:** 3
**Contribution:** 3
**Rating:** 6
**Confidence:** 4

**Summary:**

This paper proposes ViTAR (Visual Thinking and Action-centric Reasoning), a medical vision-language model that mimics clinicians’ iterative diagnostic process through a “think-act-rethink-answer” reasoning framework. Unlike conventional single-pass models, it employs a two-stage training strategy: supervised fine-tuning to guide expert-style cognitive trajectories, followed by reinforcement learning with accuracy and format rewards to optimize autonomous decision-making. To support this paradigm, the authors construct a 1K interactive instruction dataset encoding expert-like diagnostic behaviors and a 16K fine-grained VQA dataset derived from medical detection corpora. Experiments across seven medical VQA benchmarks show that ViTAR achieves state-of-the-art performance among open-source models.

**Strengths:**

1. The paper introduces a “think-act-rethink-answer” process inspired by real diagnostic workflows, offering a more interpretable and human-aligned reasoning paradigm for medical vision-language models.
2. ViTAR achieves state-of-the-art or competitive results on multiple medical VQA datasets, demonstrating that iterative reasoning and reinforcement learning can effectively enhance both diagnostic accuracy.

**Weaknesses:**

1. The core framework (think-act-rethink-answer) seems an extension of existing iterative reasoning paradigms used in general vision-language models, e.g., [1] [2]. While the paper adapts this to the medical domain, it lacks a clear claim for the fundamental difference from prior multi-step reasoning or self-reflection frameworks.
2. The paper claims that ViTAR follows a multi-step reasoning process, but there is no quantitative evidence showing that the model truly reasons step by step during inference. It is unclear whether correct answers result from genuine reasoning or accidental alignment despite incorrect intermediate steps. Metrics such as step-wise consistency or attention-ROI overlap would help verify the faithfulness of the claimed reasoning process.

[1] Perception Before Reasoning: Two-Stage Reinforcement Learning for Visual Reasoning in Vision-Language Models. 2025.

[2] Grounded Reinforcement Learning for Visual Reasoning. 2025.

**Questions:**

Please refer to the Weaknesses.

---

> ### Author Response · Authors · 2025-11-22
> **Question Response**
>
> ### **Response to Q1: Clarifying the fundamental difference from prior multi-step or self-reflective reasoning frameworks, e.g., \[1]\[2]**.
>
>
> We clarify that our ViTAR differs fundamentally from two aspects in the existing multi-step reasoning and self-reflective frameworks.
>
> 1. **Differences from general-domain iterative reasoning.**
> Existing multi-step or self-reflective reasoning frameworks are fundamentally text-driven. Their iterative loops operate on linguistic representations, while the visual perception remains static. The image is treated as a fixed input that does not change across reasoning rounds. The emerging “thinking with images” paradigm  begins to transform images from passive inputs into more interactive cognitive workspaces, but these approaches rely heavily on external tools, APIs, or domain-specific interfaces to manipulate visual content. In contrast, ViTAR learns end-to-end visual actions, predicting coordinates, marking regions, and re-examining the updated visual input. This design creates a qualitatively different reasoning process: the model actively modifies and interacts with its visual workspace through actions learned via SFT and multi-turn RL, rather than relying on text-only reflection or tool-mediated image operations.
>
> 2. **Differences from PeBR-R1[1] and ViGoRL[2].**
> PeBR-R1 produces single-round outputs and does not support visual interactive. ViGoRL achieves multi-step RL via complex multi-stage training (MCTS, tree-search linearization, SFT and RL). ViTAR achieves genuine multi-turn interactive reasoning using a much simpler SFT + RL pipeline to learn intrinsic visual actions. To the best of our knowledge, our work introduces a visual interaction formulation in multi-turn RL for multi-modal medical diagnosis, enabling explicit region localization followed by reasoning on the updated visual input. Moreover, our work provides mechanistic insights into how attention grounding and visual token allocation improve multimodal reasoning, which could inform both medical and general-domain interactive VLMs. We have added the comparison to PeBR-R1 and ViGoRL to Related work Sec. 2.2.
>
> ### **Response to Q2: Providing quantitative evidence that ViTAR performs faithful multi-step reasoning rather than accidental alignment.**
>
> We provide key evidence showing that ViTAR performs consistent step-wise reasoning and attends to semantically meaningful regions during inference.
>
> 1. **Step-wise Reasoning Consistency.**
> We use GPT-5 to evaluate the semantic consistency between the responses from think and rethink stages in the “think–act–rethink–answer” cycle. **ViTAR achieves 98.6\% step-wise consistency**, indicating that intermediate reasoning steps are stable, non-contradictory, and unlikely to result from random or accidental alignment.
>
> 2. **Attention–ROI Overlap (Faithful Visual Grounding).**
> To assess whether the model grounds its reasoning in clinically meaningful areas, we compute visual attention allocations inside vs. outside annotated ROIs. For ViTAR, **the average attention score inside ROIs is 0.2576, while the score outside ROIs drops to 0.1237**. **68.9\% of the samples exhibit higher average attention inside the ROI than outside**, showing that the model’s spatial focus aligns well with medically relevant areas. In contrast, for the baseline model, the average attention score inside ROIs is only 0.1033, slightly higher than the outside score (0.08628), and the proportion of samples with higher in-ROI attention is just 33.6\%.
>
> | Model    | Avg. In-ROI | Avg. Out-ROI | % Samples w/ Higher In-ROI |
> |----------|-------------|--------------|-----------------------------|
> | Baseline | 0.1033      | 0.0863       | 33.6%                       |
> | ViTAR    | 0.2576      | 0.1237       | 68.9%                       |
>
>
> These results collectively provide strong evidence that ViTAR’s multi-step reasoning is faithful rather than accidental, and that the model’s attention is meaningfully guided by clinically relevant regions. The results are presented as a new Table 2 and Result section "Step-wise Reasoning Consistency and Attention–ROI Overlap" in the manuscript.
>
> [1] Perception Before Reasoning: Two-Stage Reinforcement Learning for Visual Reasoning in Vision-Language Models. 2025.
>
> [2] Grounded Reinforcement Learning for Visual Reasoning. 2025.

---

### Author Response · Authors · 2025-11-27
**Inquiry Regarding Reviewer Follow-Up**

Dear PC, SAC and AC,

We have submitted our author rebuttal and have carefully addressed reviewer comments. However, we have not yet received any further responses from the reviewers.

Since the December 2 deadline is approaching, we would like to kindly ask whether our rebuttal has adequately resolved the reviewers' concerns. If there are any remaining questions or points that need clarification, we would be very happy to provide additional information or further revisions.

Thank you very much for your time and support.

Best regards,

Authors

---

### Meta-Review · Area_Chair_ZD3v · 2026-01-02

**Summary:**

The core ideas, including the iterative think–act reasoning, two-stage SFT+RL training, and reward design, largely overlap with prior work. Reviewers also questioned overstated claims of alignment with real clinical workflows due to the limited action space.

**Reviewer Concerns:**

The main outstanding concern is insufficient novelty. The proposed framework largely builds on existing iterative reasoning and SFT+RL paradigms, and the paper does not yet demonstrate any technical insights beyond an adaptation to the medical domain. Additionally, concerns remain about overstated claims of clinical workflow alignment given the limited action space, as well as unresolved questions regarding comparison fairness.

**Reviewer Scores:**

Reviewer SfZX would likely have maintained their original score, as indicated in their post-rebuttal response. Other reviewers thus would not have increased their score and would likely have maintained or reinforced their rejection.

---

### Decision · Program_Chairs · 2026-01-26

Reject